# Synaptobrevin2 monomers and dimers differentially engage to regulate the functional trans-SNARE assembly

Swapnali S Patil\* , Kinjal Sanghrajka\*, Malavika Sriram , Aritra Chakraborty, Sougata Majumdar, Bhavya R Bhaskar, Debasis Das

The precise cell-to-cell communication relies on SNARE-catalyzed membrane fusion. Among ~70 copies of synaptobrevin2 (syb2) in synaptic vesicles, only ~3 copies are sufficient to facilitate the fusion process at the presynaptic terminal. It is unclear what dictates the number of SNARE complexes that constitute the fusion pore assembly. The structure–function relation of these dynamic pores is also unknown. Here, we demonstrate that syb2 monomers and dimers differentially engage in regulating the trans-SNARE assembly during membrane fusion. The differential recruitment of two syb2 structures at the membrane fusion site has consequences in regulating individual nascent fusion pore properties. We have identified a few syb2 transmembrane domain residues that control monomer/dimer conversion. Overall, our study indicates that syb2 monomers and dimers are differentially recruited at the release sites for regulating membrane fusion events.

## Introduction

Membrane fusion is a critical cellular event that controls the secretion of chemical messengers such as neurotransmitters, hormones, cytokines, peptides (Sollner et al, 1993; Ferro-Novick & Jahn, 1994). The soluble N-ethylmaleimide–sensitive factor attachment protein receptor (SNARE) proteins catalyze secretory vesicles' fusion with the plasma membrane and serve as the minimal machinery for membrane fusion (Weber et al, 1998; McNew et al, 2000). The fusion process is divided into several mechanistically distinct steps—docking, priming, membrane merging, and recycling of fusion machinery (Sudhof, 2013). One of the key kinetic intermediates in this process is the ephemeral fusion pore, the first aqueous connection between the lumen of secretory vesicles and the extracellular space (Breckenridge & Almers, 1987; Lindau & Almers, 1995; Sharma & Lindau, 2018). The chemical messengers escape from the cell through this pathway to mediate intercellular communication. The mechanism of membrane fusion is still unclear because we do not know how the SNAREs (v-vesicular and t-target membrane SNAREs) assemble to organize the fusion pore assembly at the release site.

The SNAREs along with the membrane lipids are sufficient to catalyze fusion pore formation (Bao et al, 2016; Sharma & Lindau, 2018). Presumably, fusion pores are composed of lipids and SNARE transmembrane domains (TMDs) (Zick et al, 2014; Chang et al, 2015, 2016; Bao et al, 2016). A large number of regulatory factors directly act on SNAREs and/or membrane lipids, thereby controlling the pore opening (Chapman, 2008; Jahn & Fasshauer, 2012; Sudhof, 2013). The initial open-state duration of the pore is in the order of milliseconds; the pore then either closes (kiss-and-run exocytosis) or dilates to result in full fusion (Ceccarelli et al, 1973; Heuser & Reese, 1973; Breckenridge & Almers, 1987; Lindau & Almers, 1995). This transient nature has kept the fusion pore structure obscured till now. It is unclear how SNARE TMDs and membrane lipids contribute to the dynamic fusion pore assembly.

The purified v-SNARE synaptobrevin2 (syb2) exists as monomers and dimers in the absence of a membrane environment (Laage & Langosch, 1997; Wittig et al, 2019) and in the reconstituted liposome (Margittai et al, 1999). The syb2 (~70 copies) present in synaptic vesicles (SVs) (Takamori et al, 2006) also exists as monomers and dimers, as observed in cross-linking studies (Calakos & Scheller, 1994; Laage & Langosch, 1997). The mutagenesis studies with purified proteins identified the key TMD residues accountable for dimer formation (Laage & Langosch, 1997; Fleming & Engelman, 2001; Han et al, 2015). Two of these TMD residues when mutated altered fusion pore flux in chromaffin cells (Chang et al, 2015), but the syb2 dimerization status was not inspected in that study. The $Ca^{2+}$-triggered exocytosis in adrenal chromaffin cells was affected when syb2 TMD residues were mutated (Dhara et al, 2016); however, the effect of these mutations on syb2 dimerization was not tested. The question remains how the syb2 monomers and dimers organize into the functional SNARE complex assembly at the release site.

One of the technical difficulties in studying fusion pores has been to trap the pores in their non-dilated functional state and study the factors regulating the pores' assembly. In recent studies,

Department of Biological Sciences, Tata Institute of Fundamental Research, Mumbai, India

Correspondence: debasis.das@tifr.res.in
\*Swapnali S Patil and Kinjal Sanghrajka contributed equally to this work

fusion pores were trapped in the non-dilated state when syb2-reconstituted nanodisks (NDs) were allowed to react with the t-SNARE–reconstituted liposomes or black lipid membrane (BLM) (Shi et al, 2012; Bao et al, 2018; Das et al, 2020; Nellikka et al, 2021). Syb2 TMD mutations significantly altered the glutamate efflux through these pores (Bao et al, 2016). Syb2 copy numbers also influenced the size and kinetic properties of individual nascent pores (Bao et al, 2018). Here, we have investigated the role of syb2 monomers and dimers in shaping the functional SNARE complex assembly at the membrane fusion site. Our ensemble biochemical studies using SVs and other reconstituted membranes are supported by high-resolution single-pore measurements and studies with live PC12 cells.

The ratio of syb2 monomers to dimers, both in SVs and in reconstituted membranes, serves as functional components at the onset of the trans-SNARE complex formation. This ratio appears to be heterogeneous under physiological conditions. Our in vitro experiments revealed that the membrane lipid composition and the presence of additional membrane proteins regulate the monomer-to-dimer ratio in the donor membrane. When donor membrane–resident syb2 encountered t-SNAREs in the acceptor membrane, both its monomer and dimer population contributed to the trans-SNARE assembly in a t-SNARE density-dependent manner. The varying abundance of these syb2 structures in the donor membrane has ramifications in the fusion pore assembly. The syb2 TMD residues involved in controlling the dimer-to-monomer ratio also regulate the stability and size of individual nascent fusion pores. The dimers express abundantly in the plasma membrane of PC12 cells during stimulated secretion. Overall, this study showed the role of syb2 monomers and dimers in differentially regulating membrane fusion events at the release sites.

## Results

### Syb2 monomers and dimers differentially populate the trans-SNARE assembly at the onset of membrane fusion

We first probed the occurrence of syb2 monomers and dimers in the physiological environment (Fig 1A). SVs isolated from adult rat brains (Fig S1A–C) showed a detectable syb2 dimer population (Fig 1A), which was significantly less compared with the monomers (Fig 1A). The addition of cross-linker DSP (dithiobis succinimidyl propionate) enhanced the dimer population in SVs (Fig S1D), also shown previously (Calakos & Scheller, 1994; Edelmann et al, 1995). It indicates that syb2 monomers are indeed present nearby within SVs; however, membrane lipids and/or SV-resident protein(s) presumably prevent syb2 dimerization. Interestingly, the dimer population exists significantly more in synaptosome fractions, in comparison with SVs, resulting in a significant increase in the dimer-to-monomer ratio (D/M) in synaptosome fractions than in SVs (Fig 1A). This set of experiments indicated the existence of a heterogeneous syb2 D/M in the physiological environment; however, the functional consequence of this heterogeneity was unclear.

To get additional molecular insight, we reconstituted syb2 in different donor membranes. An immunoblot showed the presence of both the monomers and the dimers in purified protein syb2 (syb2), syb2 reconstituted in liposomes (v-SNARE liposomes, v-lipos), and NDs (~five copies of WT syb2 reconstituted in NDs with ~13 nm diameter—ND5$_S$) (Bao et al, 2018; Nellikka et al, 2021) (Figs 1B and S2A and B). To confirm whether the syb2 dimer band in the immunoblots are indeed homodimers, we performed mass spectrometry analysis of syb2 dimers and monomers using ND5$_S$ (Fig S3). The syb2 dimers in ND5$_S$ samples were populated by the additional peaks corresponding to MSP (membrane scaffolding protein, MSPE3D1) peptides (Fig S3).

Then, we investigated the extent to which the syb2 monomers and dimers engage in the trans-SNARE assembly. The v-lipos, ND5$_S$, and SVs were separately allowed to react with the liposomes containing ~200 copies of t-SNAREs (Fig 1C). The SNARE complex formation was traced in the immunoblots under all these conditions by probing either syb2 or SNAP-25B (Figs 1D–F and S4A–C) (Tolar & Pallanck, 1998; Chen et al, 1999; Nellikka et al, 2021). During the SNARE assembly, band intensity corresponding to syb2 dimers reduced significantly unlike monomers, as shown in the immunoblots (Figs 1D–F and S4A–C). We quantitatively estimated the dimers and monomers, in the absence and presence of t-SNARE liposomes (t-lipos). At this t-SNARE density, the normalized band intensity showed a significant reduction in the dimer population, whereas the reduction for monomers and other higher order syb2 oligomers was not significant (Figs 1D–F and S5). It was unclear whether the syb2 dimers' disappearance indicates whether they dissociated into monomers or were recruited in the SNARE complex assembly.

To further verify, we site-specifically labeled syb2 TMD residue C103 with the fluorescent dye cy3 or cy5, in different preparations (Fig 2A–D). These two fluorescent dyes serve as the FRET pair, also reported earlier (Das et al, 2020). The two fluorophores labeled syb2 were incorporated together in different donor membranes—v-lipo to yield v-lipo$^{cy3/cy5}$ and ND to yield ND5$_S^{cy3/cy5}$. Two different sets of experiments with these donor membranes were designed. At first, when v-lipo$^{cy3/cy5}$ was allowed to react with t-lipos, an increase in the cy5 emission was observed with time as we excited the samples at cy3 excitation wavelength (550 nm) (Fig 2B and C). This increase was t-lipo concentration–dependent (Fig 2B and C), indicating that the syb2 dimers assemble in the presence of t-lipo as the trans-SNARE complexes are formed. Presumably, the disappearance of syb2 dimers during the SNARE assembly, as shown above, was due to dimers' differential engagement with the SNARE complexes.

To further verify, we used ND5$_S^{cy3/cy5}$ as the donor membrane and allowed it to react with the t-lipo (Fig 2D). We resolved those samples in SDS–PAGE and traced the cy3/cy5-labeled syb2 using a fluorescence imager (Fig 2D). Interestingly, the syb2 dimers populated the SNARE complex assembly, as revealed by imaging at cy5 emission wavelength after exciting the samples at cy3 excitation wavelength (Fig 2D).

This set of experiments suggested that the syb2 monomer-to-dimer ratio is crucial in differentially regulating trans-SNARE complex formation.

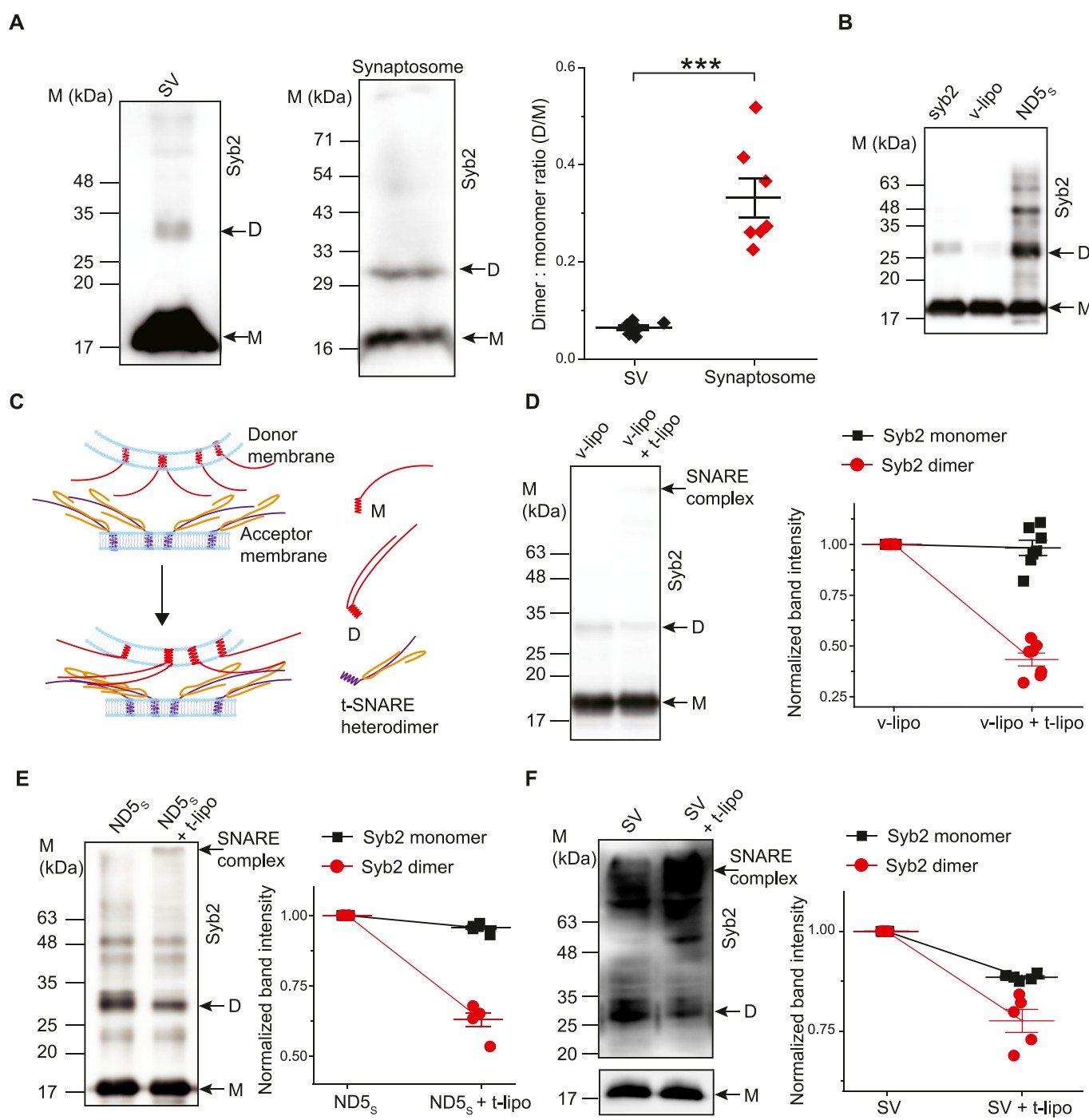

**Figure 1.   Syb2 dimer level in the donor membrane reduces during the SNARE complex assembly.**
**(A)** Left—representative immunoblots showing the presence of syb2 monomers (M) and dimers (D) in synaptic vesicles (SVs) isolated from rat brains. The antibody used is mentioned. n = 10 independent blots; N = 4 independent SV isolation. Middle—representative immunoblots showing the presence of syb2 monomers (M) and dimers (D) in synaptosome fractions isolated from rat brains. The antibody used is mentioned. n = 7 independent blots; N = 3 independent synaptosome isolation. Right—dimer-to-monomer band intensity ratio is plotted as "Dimer: monomer ratio" corresponding to SV (black) and synaptosome (red). n = 7 independent blots; N = 3 independent isolation; individual data points are shown along with the mean ± SEM. The *t* test was performed to compare the two means, ***$P < 0.001$. **(B)** Representative immunoblot showing the presence of syb2 monomers (M) and dimers (D) in the purified protein syb2 and syb2 present in the reconstituted membranes—v-lipo and nanodisks (ND5$_S$). The antibody used is mentioned. n = 3 independent experiments. **(C)** Illustration shows the involvement of syb2 monomers (M) and dimers (D) in the trans-SNARE assembly. **(D)** Left panel—representative immunoblots showing syb2 monomers (M) and dimers (D) in v-lipos in the absence (v-lipo) and presence (v-lipo + t-lipo) of t-lipos. SNARE complex formation is indicated. Right panel, band intensities corresponding to v-SNARE plus t-lipos (v-lipo + t-lipo) have been normalized to v-lipos, for syb2 monomers (black) and dimers (red). n = 6 independent experiments; individual data points are shown along with the mean ± SEM; N = 2 different liposome preparations. **(E)** Left panel—representative immunoblots showing syb2 monomers (M) and dimers (D) in v-SNARE NDs in the absence (ND5$_S$) and presence (ND5$_S$ + t-lipo) of t-lipos. SNARE complex formation is indicated. Right panel, band intensities corresponding to ND5$_S$ plus t-lipos (ND5$_S$ + t-lipo) have been normalized to ND5$_S$, for syb2

## The t-SNARE availability dictates recruitment of syb2 monomers and dimers in the SNARE assembly

Because t-SNAREs are not homogeneously distributed in the plasma membrane (Rickman et al, 2010; van den Bogaart et al, 2011; Sharma & Lindau, 2017), it is unclear how many t-SNAREs engage during the functional fusion pore assembly. Next, we investigated the role of t-SNARE abundance in regulating syb2 monomers/dimers recruitment at the release site. To test this, we first allowed ND5$_S$ to react with liposomes bearing increasing t-SNARE copies (from 25 to 200, yielding t-lipo$_{25}$, t-lipo$_{50}$, t-lipo$_{100}$, and t-lipo$_{200}$); however, the number of t-lipos was kept constant to the number of NDs. Interestingly, syb2 dimers' alteration was less prominent in the presence of t-lipo$_{25}$ or t-lipo$_{50}$, whereas the dimers altered significantly when t-lipo$_{100}$ or t-lipo$_{200}$ engaged in the SNARE complex assembly (Fig 3A and B). The syb2 monomers reduced as t-lipo$_{25}$ or t-lipo$_{50}$ were used in different sets of experiments, but no additional reduction was observed in the presence of t-lipo$_{100}$ or t-lipo$_{200}$ (Fig 3A and B).

Then, we allowed SVs to react with an increasing number of t-lipo, keeping the t-SNARE copies constant (~200) in the liposomes. Interestingly, upon an increase in the number of t-lipos, band intensity in the immunoblots corresponding to syb2 monomers also started to decrease along with the dimers (Fig 3C and D). In all these conditions, SV-resident proteins synaptophysin (syp) and synaptotagmin1 (syt1) were also probed (Fig 3C and D). The syp and syt1 levels did not alter significantly (Fig 3C and D).

Hence, t-SNARE abundance in the acceptor membrane differentially controls syb2 monomer/dimer stoichiometry during the SNARE complex assembly. It was, however, unclear whether syb2 distribution in the donor membrane also impacts the functional trans-SNARE assembly at the release site.

## Syb2 availability and distribution influence the dimer/monomer ratio in the donor membrane

The SV contains ~70 copies of syb2 (Takamori et al, 2006), whereas only ~2–3 copies are sufficient to elicit membrane fusion (Sinha et al, 2011; Bao et al, 2018). To investigate whether syb2's availability influences the dimer-to-monomer ratio (D/M) in the donor membrane, we first incorporated three (ND3$_S$), five (ND5$_S$), and seven (ND7$_S$) copies of syb2 in ~13 nm NDs (Fig S6). In the immunoblot, the band intensity corresponding to the dimer population increased linearly from ND3$_S$ to ND7$_S$, leading to an increased D/M (Fig 4A–C). When t-lipos were treated with these sets of NDs, D/M decreased for all the NDs tested—ND3$_S$, ND5$_S$, and ND7$_S$ (Fig 4A–C).

To get additional insight, we then altered the syb2 density in NDs by reconstituting its five copies in larger size NDs (~50 nm in diameter, Fig S7A and B) (Nasr et al, 2017; Bao et al, 2018). That yielded ND5$_L$, which possesses low syb2 densities compared with ND5$_S$ (Fig 4D–F). ND5$_L$ showed a significant reduction in D/M in comparison with ND5$_S$, as quantified from the immunoblots (Fig 4D–F). Presumably, the dilution of syb2 in larger NDs (ND5$_L$) did not allow enough syb2 dimer formation. We increased the syb2 density of ND5$_L$ yielding ND55$_L$ (Fig S7), which showed an increased D/M compared with ND5$_L$ (Fig 4D–F). Notably, in the case of ND5$_L$, dimer population reduction was observed during the SNARE complex assembly (Fig S8A and B). The results indicated that the spatial arrangement of syb2 in the donor membrane controls the abundance of monomers and dimers.

To trace the potential factors that can control the syb2 abundance, we then compared syb2 monomers and dimers within NDs, after varying its membrane lipid composition (Fig 4G–I). The D/M reduced significantly when the negatively charged lipids were excluded from NDs (Fig 4G and H). The presence of the neutral membrane lipid also altered the glutamate release through the fusion pores, when glutamate-entrapped t-SNARE vesicles were allowed to react with ND5$_S$ (with varying lipid compositions) (Fig S9A and B). Next, we varied the cholesterol content of ND5$_S$ and found that D/M increased significantly when the membrane lipids contained cholesterol in physiological concentration (Fig 4I). In the case of SVs, the cholesterol scavenger M$\beta$CD (methyl-$\beta$-cyclodextrin) (Wu et al, 2021) treatment considerably altered the syb2 dimer population (Fig 4J), indicating membrane lipid's role in controlling D/M within SV (Tong et al, 2009). In the reconstituted membrane, we included SV-resident protein syt1 in addition to syb2 within NDs and checked the syb2's dimerization status. The syt1's presence also reduced D/M within NDs (Fig 4K). This set of experiments suggests that the syb2 monomers and dimers are dynamic and their abundance is regulated by the membrane lipids and/or SV-resident membrane proteins. It was, however, unclear whether syb2 monomers' and dimers' varying abundance impacts the functional fusion pore assembly.

## Syb2 dimer/monomer ratio affects the functional fusion pore assembly at the onset of fusion

Because the factors demonstrated above influenced syb2 D/M, both in reconstituted membranes and in SVs, we wondered how that would impact the fusion pore properties. To investigate that, we used a recently described ND-BLM assay (Bao et al, 2018; Das et al, 2020; Nellikka et al, 2021) and studied the sub-millisecond dynamics of individual nascent fusion pores (Fig 5A). The BLM containing t-SNAREs was treated with NDs harboring varying syb2 copies in different sets of experiments (Fig S6). Notably, ND3$_S$ yielded pores with significantly different sizes and kinetic properties in comparison with ND5$_S$ or ND7$_S$ (Bao et al, 2018). ND3$_S$ also showed the least syb2 D/M among these NDs (Fig 4A–C). To probe

---

monomers (black) and dimers (red). n = 5 independent experiments; individual data points are shown along with the mean ± SEM; N = 3 different liposome preparations.
**(F)** Left panel—representative immunoblots showing syb2 monomers (M) and dimers (D) in SVs in the absence (SV) and presence (SV + t-lipo) of t-lipos. SNARE complex formation is indicated. Right panel, band intensities corresponding to SV plus t-lipos (SV + t-lipo) have been normalized to SV, for syb2 monomers (black) and dimers (red). n = 4 independent experiments; individual data points are shown along with the mean ± SEM; N = 2 different SV preparations. The antibodies used are mentioned for all the immunoblots.
Source data are available for this figure.

**Figure 2. Syb2 dimers engage in the SNARE complex assembly.**
**(A)** Illustration shows the increase in fluorescently labeled syb2 dimer (D) population, during the SNARE assembly in the presence of t-SNARE heterodimers (T), as indicated. The syb2 monomers (M), labeled at C103 position in TMD with either cy3 (green) or cy5 (blue) fluorescent dyes, are indicated. **(B)** Representative data showing the change in fluorescence intensity with time, when t-lipo reacts with fluorescently labeled syb2 in v-lipo - v-lipo$^{cy3/cy5}$ (v-lipo$^{cy3/cy5}$:t-lipo—1:1 for black square, 1:6 for red circle). n = 4 independent experiments. **(C)** Pooled data from (B), showing fold change in maximum fluorescence intensity relative to v-lipo$^{cy3/cy5}$ alone; n = 4 independent experiments; individual data points are shown along with the mean ± SEM. **(D)** Reaction between ND5$_S$$^{cy3/cy5}$ (fluorescently labeled syb2 reconstituted in NDs) and t-lipo was resolved in SDS–PAGE and developed under fluorescence imager (excitation and emission wavelengths are 532 and 670 nm, respectively). Bands corresponding to SNARE complexes and the syb2 dimer (D) are indicated. n = 3 independent experiments.
Source data are available for this figure.

the impact of D/M on individual pore properties, we compared the pores formed by ND5$_S$, ND5$_L$ and ND55$_L$ (Fig 5A). The t-SNAREs present in the BLM and the membrane lipid compositions were the same in all sets of experiments. Interestingly, ND5$_L$ yielded pores with a noticeably smaller diameter than the ND5$_S$, evident from the current measurements (Fig 5A–C). In addition, ND5$_L$ pores' open states were significantly destabilized; the mean open-state dwell time ($<t_{o-obs}>$) was 2.4 (±0.03) ms in the case of ND5$_L$, which was ~450 times shorter than the ND5$_S$ pores (1,132.5 [±115.1] ms). The open-state dwell time distribution also shifted drastically to the shorter open lifetimes for ND5$_L$ in comparison with ND5$_S$ (Fig 5D). When the syb2 density was restored in ND55$_L$, the pore diameter and the open-state stability enhanced significantly compared with ND5$_L$ (Fig 5A–D). Because syb2 D/M was dependent on its density in the

ND preparations (Fig 4D–F), the spatial arrangement of these syb2 structures was presumably responsible for the altered pore properties. These properties control the size and open-state stability of the individual nascent fusion pores.

### Syb2 TMD residues that control D/M impact pores' properties

Next, we aimed to identify specific TMD residues responsible for syb2 D/M in the donor membrane, which contributes to fusion pores' size and kinetic stability. We first screened a series of syb2 TMD mutations for their ability to control D/M in a membrane-reconstituted system and identified a few responsible TMD residues—V101 and I105 (Fig 6A and B and S10A–C). ND5$_S$ (V101A) and ND5$_S$(I105A) mutations

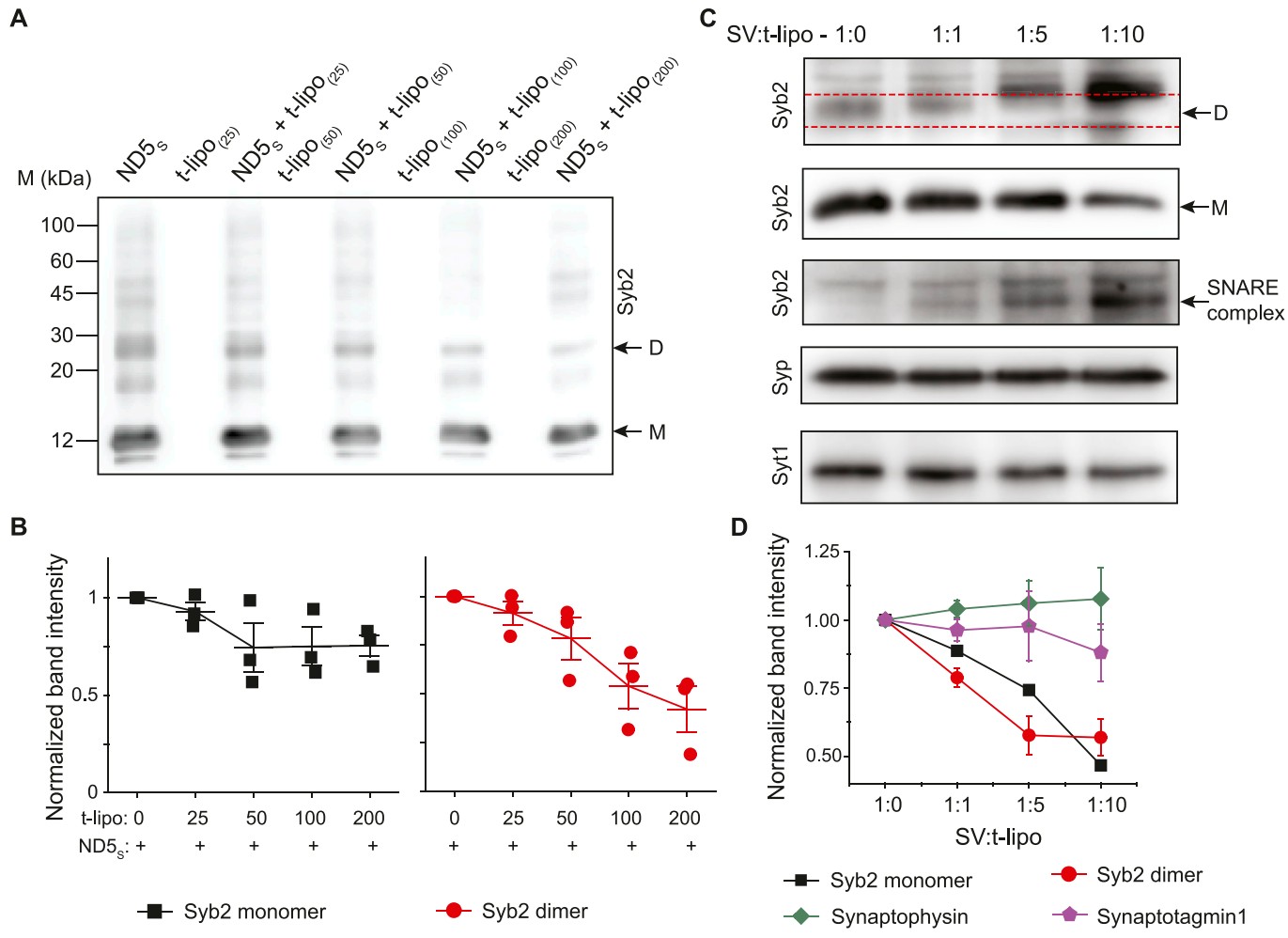

**Figure 3. Syb2 monomer and dimer engagement in the SNARE assembly depends on the acceptor membrane's t-SNARE abundance.**
**(A)** Representative immunoblots showing syb2 monomers (M) and dimers (D) in $ND5_S$ in the absence and presence of t-lipos, harboring varying t-SNARE copies as indicated. The antibody used is mentioned. n = 3 independent experiments. **(A, B)** Band intensities corresponding to monomers (black, left panel) and dimers (red, right panel) in the presence of t-lipos (as described in (A)) are plotted after normalizing with the corresponding band intensities of $ND5_S$ alone (t-lipo: 0). n = 3 independent experiments; individual data points are shown along with the mean ± SEM. **(C)** Representative immunoblots showing syb2 monomers (M), dimers (D), syp, and syt1 in SVs, before and after treating with t-lipos. Here, the number of SVs was kept constant, whereas the number of t-lipos was varied, as indicated by the ratios. SNARE complex formation is indicated. The antibodies used are mentioned. n = 4 independent experiments; N = 2 different SV preparations. **(D)** Band intensity corresponding to monomers (black), dimers (red), syp (green), and syt1 (purple) in the presence of t-lipos (as described in (C)) is plotted after normalizing with the corresponding band intensities of SV alone (SV: t-lipo—1:0). n = 4 independent experiments; data are represented as the mean ± SEM.
Source data are available for this figure.

reduced the syb2 D/M compared with WT (Fig 6B and C). Because most of the TMD isoleucine (I) residues are involved in dimerization (Laage & Langosch, 1997; Laage et al, 2000; Han et al, 2015), we also generated a syb2 mutant All ItoA, where all TMD isoleucine (I) residues were mutated to alanine (A) residues (Fig 6A). $ND5_S$(All ItoA) showed a marked reduction in the syb2 D/M, and the reduction was more compared with the other two mutations (Fig 6B and C). These studies demonstrated the importance of syb2 TMD residues in regulating the stability of monomers and dimers in the donor membrane.

During the functional SNARE complex assembly, $ND5_S$ bearing the above syb2 mutations were allowed to react with t-lipo in different sets of experiments. These syb2 mutants were capable of forming the trans-SNARE assembly as the syb2 D/M reduced

significantly (Fig 6B and C) during complex formation (Fig S11). The lipid mixing assay, described previously (Weber et al, 1998; Shi et al, 2012), was used to quantify the effect of these mutations in membrane fusion (Fig 6D and E). The quantified $t_{1/2}$ for lipid mixing increased significantly for all the mutations in comparison with WT (Fig 6D and E). It was, however, unclear whether these mutations directly affect fusion pore properties.

All the syb2 mutations yielded pores with a comparable size to that of WT syb2, indicated by the current histograms (Fig S12). $ND5_S$(All ItoA) pores, however, showed significantly more subconductance states compared with $ND5_S$ (Figs 6F and S12A). $ND5_S$(V101A) also showed substantial destabilization of the pore open states, as observed from the raw traces and the current histograms (Figs 6F and S12B). $ND5_S$(I105A) also altered the pore

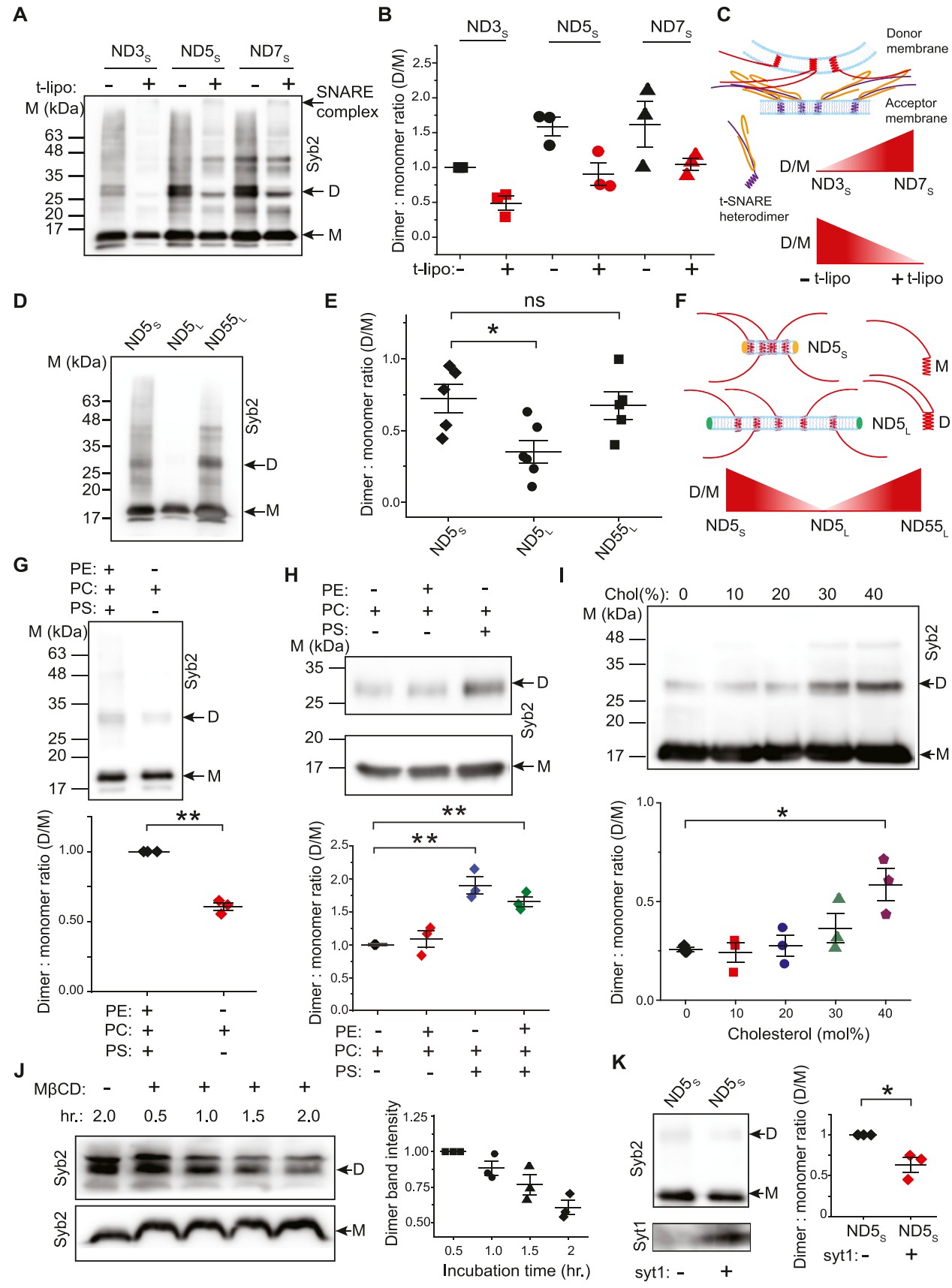

properties more distinctly than the above two mutations (Figs 6F and S12B). The open-state dwell time distribution of pores with syb2 TMD mutations also shifted significantly to the shorter open lifetime in comparison with ND5$_S$ pores (Fig 6G). The <$t_{o-obs}$> of ND5$_S$(All ItoA), ND5$_S$(V101A), and ND5$_S$(I105A) pores were 36 (±1) ms, 301 (±13) ms, and 811 (±89) ms, respectively, which are significantly shorter than the <$t_{o-obs}$> of ND5$_S$ pores (1,132.5 [±115.1] ms). The V-to-A or I-to-A mutations in the syb2 TMD can decrease the van der Waals interaction between the membrane and the TMD region of syb2 and can contribute to altering the pore properties. To check that, we have used syb2 L99W and C103W mutants; this will increase the above van der Waals interaction. These L99W and C103W mutants did not affect syb2 dimer population significantly. As shown in Fig S13, none of these mutants altered the size or open-state kinetic stability of individual pores, as evident from the current histograms and the open-state dwell time distribution (Fig S13A–C). Hence, the van der Waals interaction did not affect the fusion pore properties in the case of V-to-A or I-to-A mutations in the syb2.

The above data show the relative contribution of specific syb2 TMD residues to differentially altering D/M, which has ramifications in individual pore properties.

### Syb2 dimers abundantly express in the plasma membrane of PC12 cells during secretion

Next, we checked whether syb2 dimers express in the living cell, which is critical in controlling syb2 D/M. To test that, a BiFC (bimolecular fluorescence complementation) assay (Hu et al, 2002; Kerppola, 2006) was performed. We transfected PC12 cells with a plasmid containing two syb2 molecules, each fused with one part of the split mCherry (Fdez et al, 2010; Feng et al, 2019) (syb2/split-mCherry) (Figs 7A and B and S14A). The mCherry expression inside the cell indicated the presence of syb2 dimers (Fig 7C). When we

removed one half of the split mCherry from the plasmid (syb2/half-mCherry), no significant mCherry signal was detected (Figs 7C and S14B), further confirming that the mCherry signal was indeed originating from syb2 dimer population expressing inside the PC12 cells.

To investigate whether syb2 dimers localize at the plasma membrane, we first marked the plasma membrane boundary of the syb2/split-mCherry–expressing PC12 cells by exogenously adding FM1-43 dye (Klima & Foissner, 2008) (Fig 7D). We investigated how fluorescence signals from FM1-43 and mCherry alter with time, before and after KCl stimulation. The contribution of photobleaching was included in the quantification (Fig S15A and B). We observed a gradual decrease in FM1-43 signal upon KCl stimulation (Fig 7D), an indication of vesicular secretion from PC12 cells (Gaffield & Betz, 2006). Interestingly, the corresponding mCherry signal increased briefly upon KCl stimulation followed by a decrease with time (Fig 7D). The FM1-43 signal decay followed a double exponential kinetics, with minor fast and major slow kinetic components (Table S1). The concomitant mCherry signal increase followed by a decrease suggested that new dimers accumulate at the plasma membrane upon KCl stimulation.

This set of experiments suggested the presence of syb2 dimers in the plasma membrane of PC12 cells, during secretion.

## Discussion

At present, very little is known about the structure–function relation of fusion pores, the first aqueous connection between the vesicular lumen and cell exterior. In neurons, SVs harbor ~70 copies of syb2 (Takamori et al, 2006), but only ~2–3 copies are sufficient to catalyze the fusion pore assembly at the release site (Sinha et al, 2011; Shi et al, 2012; Bao et al, 2018). The syb2 copy numbers also regulate the

**Figure 4. Syb2 dimers are dynamic in reconstituted membranes and SVs.**
**(A)** Representative immunoblots showing syb2 monomers (M) and dimers (D) in NDs reconstituted with 3 (ND3$_S$), 5 (ND5$_S$), and 7 (ND7$_S$) syb2 copies, in the absence and presence of t-lipos. SNARE complex formation is indicated. The antibody used is mentioned. n = 3 independent experiments; N = 2 different ND preparations. **(B)** Dimer-to-monomer band intensity ratio was normalized to ND3$_S$ without t-lipos (black square), and is plotted as "Dimer: monomer ratio" corresponding to ND3$_S$ with t-lipo (red square), ND5$_S$ without (black circle) and with (red circle) t-lipo, and ND7$_S$ without (black triangle) and with (red triangle) t-lipo. n = 3 independent experiments; N = 2 different ND preparations; individual data points are shown along with the mean ± SEM. **(C)** Illustration shows that the syb2 copy numbers impact D/M in the donor membrane. During the trans-SNARE assembly, syb2 dimers significantly reduce upon interaction with the t-SNAREs in the acceptor membrane. **(D)** Representative immunoblots showing syb2 monomers (M) and dimers (D) in syb2-reconstituted NDs with varying diameters, as indicated; suffixes "S" and "L" indicate small and large diameter NDs, respectively. The antibody used is mentioned. n = 5 independent experiments; N = 3 different ND preparations. **(E)** Dimer-to-monomer band intensity ratio is plotted for all NDs (as indicated). n = 5 independent experiments; N = 3 different ND preparations. The t test was performed to compare the two means, *P < 0.05 and ns > 0.05. **(F)** Illustration shows the impact of syb2 density on the donor membrane's D/M. **(G)** Top panel, representative immunoblot showing the effect of membrane lipids on syb2 D/M. Lipid composition for ND5$_S$ and the antibody used are mentioned. Bottom panel, dimer-to-monomer band intensity ratio was normalized to ND5$_S$ composed of PE, PC, and PS (black). Data for ND5$_S$ composed of PC alone are shown in red. n = 3 independent experiments; individual data points are shown along with the mean ± SEM. The t test was performed to compare the two means, **P < 0.01. **(H)** Top panel, representative immunoblot showing the effect of membrane lipids on syb2 D/M. Lipid composition for ND5$_S$ and the antibody used are mentioned. Bottom panel, dimer-to-monomer band intensity ratio was normalized to ND5$_S$ composed of PC alone (black). Data for ND5$_S$ composed of PC, PE (red), PC, PS (blue), and PC, PE, PS (green) are shown. n = 3 independent experiments; individual data points are shown along with the mean ± SEM. The t test was performed to compare the means, **P < 0.01. **(I)** Top panel, representative immunoblot showing the effect of cholesterol on syb2 D/M. Lipid composition for ND5$_S$ and the antibody used are mentioned. Bottom panel, dimer-to-monomer band intensity ratio is plotted as a function of ND5$_S$ membrane's cholesterol content; individual cholesterol (mol%) concentrations are indicated by black, red, blue, green, and purple. n = 3 independent experiments; individual data points are shown along with the mean ± SEM. The t test was performed to compare the two means, *P < 0.05. **(J)** Left panel, representative immunoblot showing syb2 monomers (M) and dimers (D) in SVs after M$\beta$CD treatment for the respective time (in hour), as indicated. Right panel, dimer band intensities for different time points were normalized to the band intensity corresponding to 0.5-h treatment, and are plotted as a function of incubation time (hr.). n = 3 independent experiments; individual data points are shown along with the mean ± SEM. N = 2 independent SV isolation. **(K)** Left panel, representative immunoblot showing syb2 monomers (M) and dimers (D) in ND5$_S$ with (+) or without (−) syt1. The total number of syb2 molecules was the same in both the ND preparations. Right panel, dimer-to-monomer band intensity ratio was normalized to ND5$_S$ without syt1 (black). Data for ND5$_S$ containing syt1 are shown in red. n = 3 independent experiments; individual data points are shown along with the mean ± SEM. The t test was performed to compare the two means, *P < 0.05.
Source data are available for this figure.

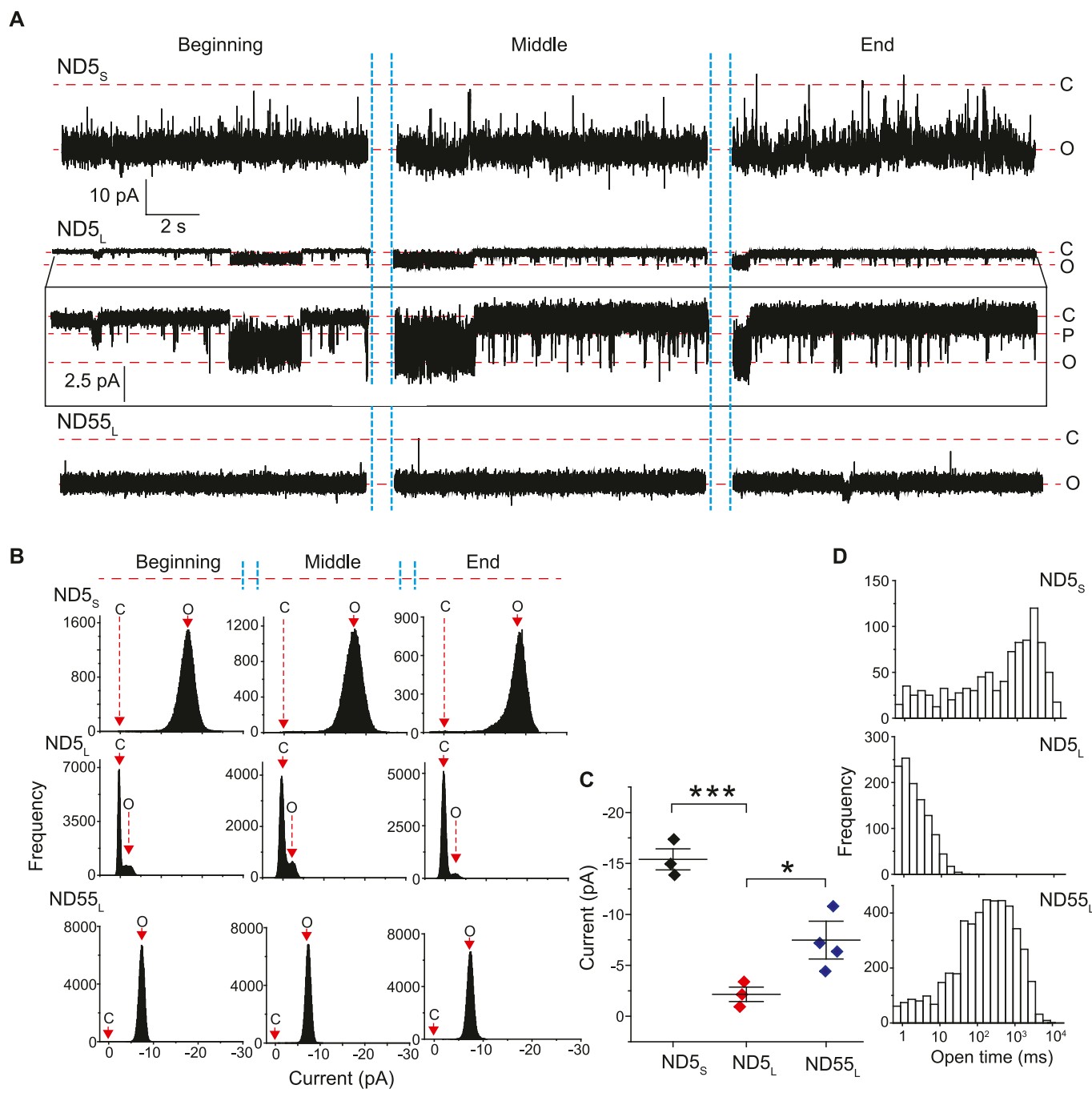

**Figure 5. Syb2 monomer and dimer abundance impacts individual fusion pore properties.**
**(A)** Representative traces of single pores for ND5$_S$ (top), ND5$_L$ (middle), and ND55$_L$ (bottom), as indicated. Three different epochs of the raw traces—beginning, middle, and end—are shown. The ND5$_L$ pore's current axis is enlarged and shown below, within the box. Closed (C), open (O), and partial open (P) states are indicated with the respective currents. **(A, B)** Current histograms corresponding to the traces shown in (A) (ND5$_S$—top; ND5$_L$—middle; and ND55$_L$—bottom). Closed (C) and open (O) states are indicated. **(C)** Individual pore currents are plotted for ND5$_S$, ND5$_L$, and ND55$_L$, mean ± SEM (n = 3 independent BLMs, each for ND5$_S$ and ND5$_L$; 4 independent BLMs for ND55$_L$; two different ND preparations were used). The t test was performed, ***P < 0.001 and *P < 0.05. **(D)** Open dwell time histograms of pores, as indicated. n = 3 independent BLMs, each for ND5$_S$ and ND5$_L$; four independent BLMs for ND55$_L$; two sets of NDs of each type were used.

pore properties (Bao et al, 2018). It is unclear how heterogeneously distributed syb2 stoichiometrically interacts with the t-SNAREs at the onset of membrane fusion. This is at the heart of delineating factors controlling membrane fusion's spatiotemporal dynamics

(Gramlich & Klyachko, 2019). The syb2 is present as monomers and dimers in the absence of a membrane environment and in SVs(Calakos & Scheller, 1994; Laage & Langosch, 1997), and its dimerization motif has been identified as the TMD (Laage & Langosch,

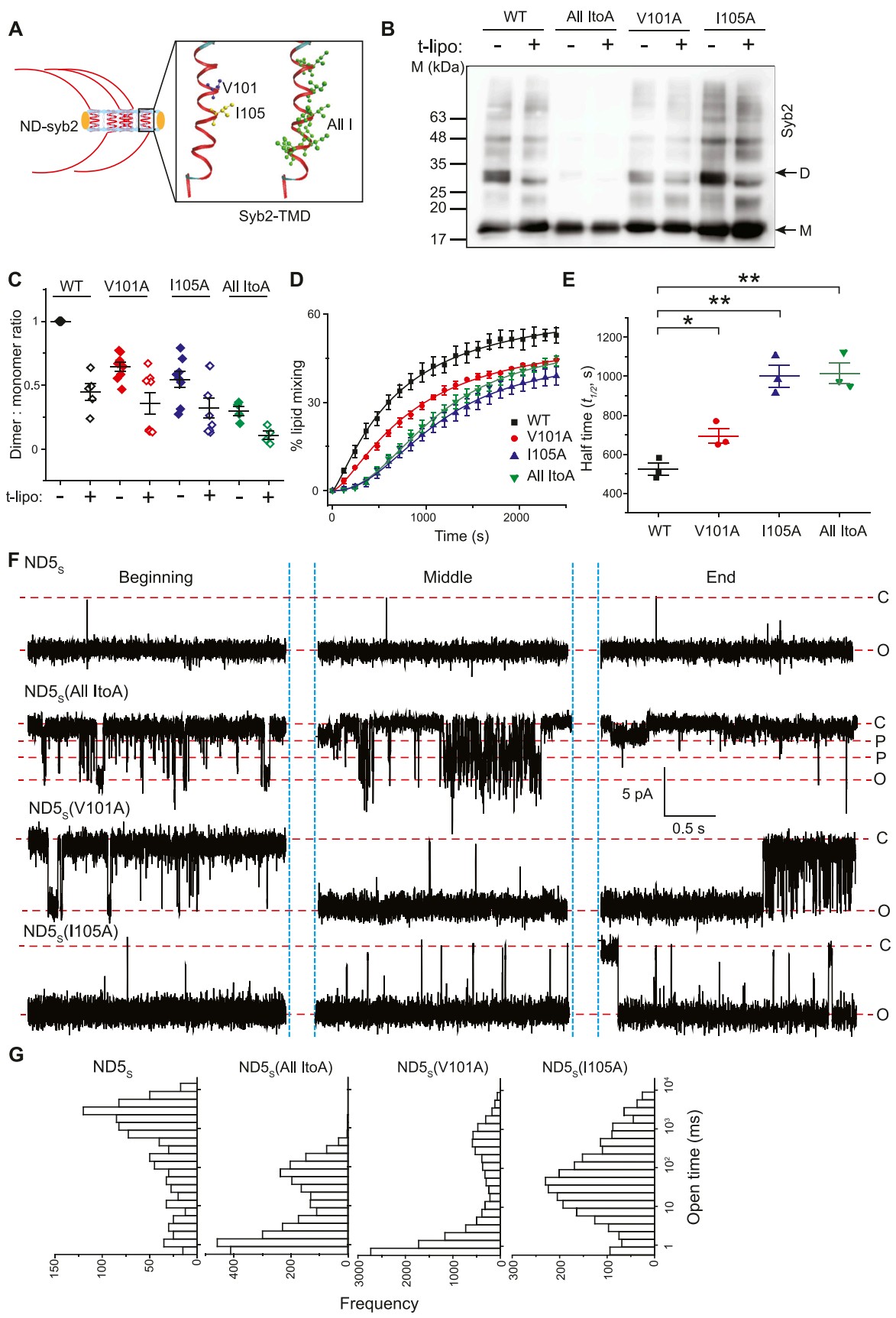

1997; Kroch & Fleming, 2006; Han et al, 2015). The syb2 TMD residues responsible for dimer formation have been identified using purified proteins in the absence of a membrane environment (Laage & Langosch, 1997; Bowen et al, 2002; Roy et al, 2004). A previous study demonstrated differential levels of syb2 monomers and dimers in SVs isolated from embryonic and adult brains (Becher et al, 1999). The functional importance of different syb2 structures in organizing the fusion pore assembly is still unknown. In our experiments, SVs harbored significantly less dimers in comparison with monomers. However, D/M increased ~fivefold in synaptosome fractions than SVs, indicating that D/M is heterogeneous in different physiological conditions.

To investigate the impact of heterogeneous syb2 D/M in membrane fusion, we checked how syb2 monomers and dimers engage in the functional SNARE complex assembly in the presence of t-SNAREs. When v-lipos, ND5$_S$, or SVs were separately engaged in the SNARE assembly, strikingly, the syb2 dimer population reduced more compared with its monomers, as observed in the immuno-blots (Fig 1D–F). The differential alteration in syb2 monomer and dimer band intensities could be due to dimer breakage into monomers or due to syb2 structures' differential engagement into the SNARE complex. When v-lipo$^{cy3/cy5}$ was allowed to react with t-lipo, the FRET read-out increased with time in a t-lipo concentration–dependent manner, indicating that multiple dimers presumably accumulate during the SNARE complex assembly (Fig 2B and C). Furthermore, the SNARE complexes formed during ND5$_S^{cy3/cy5}$ and t-lipo reaction also contained syb2 dimers, as evi-dent from the fluorescence imaging done on this set of samples (Fig 2D). In this set of imaging experiments, it was not evident whether syb2 dimer levels alter during the SNARE complex assembly, which was evident in the immunoblots. It should be noted that in this set of experiments, we could only analyze a fraction of syb2 dimers formed between cy3- and cy5-labeled syb2. We could not reliably analyze syb2 dimers formed within each of the cy3- or cy5-labeled syb2 population. Overall, our results suggested that the syb2 D/M is critical in regulating the SNARE complex assembly. How this reg-ulation affects the equilibrium between monomers and dimers is, however, unknown. The kinetic intermediates that drive the for-mation of SNARE complexes are yet to be elucidated. Additional experiments with new techniques will be required to study the dynamic behavior of syb2 monomers and dimers. In the immu-noblots, we also observed higher order syb2 oligomers (viz., trimers, tetramers, etc.). However, these higher order oligomers observed in

the immunoblots did not alter significantly during the SNARE complex assembly (Fig S5).

Next, we increased the t-SNARE abundance in our acceptor membranes and checked how syb2 dimers and monomers from SVs and NDs engage in the trans-complex formation. As expected, syb2 monomer levels along with dimers started to alter during the trans-SNARE assembly under this condition (Fig 3). The v- and t-SNARE's stoichiometry in the fusion pore assembly at the release site is unknown (Fang & Lindau, 2014; Chang et al, 2017). Previous reports also suggest that the t-SNAREs are present in clusters at the plasma membrane's active zone (Lang et al, 2001; Barg et al, 2010). Pre-sumably, syb2 monomers and dimers differentially engage in regulating the fusion pore assembly, based on the differential t-SNARE availability at the plasma membrane.

Next, we inspected the factors regulating syb2 D/M. The syb2 copy number and density in the NDs had a direct impact on D/M. When we excluded negatively charged lipids from the recon-stituted membrane, the dimer-to-monomer ratio significantly reduced (Fig 4G and H). The addition of cholesterol, however, had an opposing effect (Fig 4I). Similarly, scavenging cholesterol from the SVs had a negative impact on syb2 D/M (Fig 4J). The membrane lipid perturbations as demonstrated above might not be fully replicated inside the cell. However, the local lipid dynamics has been shown to impact membrane fusion (Churchward et al, 2008; Zhang & Jackson, 2010; Kreutzberger et al, 2017; Dhara et al, 2020). Our results indicate that the local lipid dynamics might impact the syb2 D/M, which has ramifications in the fusion pore assembly. The SVs house many membrane proteins along with syb2 (Takamori et al, 2006). When we co-reconstituted syt1 along with syb2 in the NDs (Das et al, 2020), they significantly reduced the syb2 D/M (Fig 4K). Hence, syb2 D/M in reconstituted membranes or SVs is controlled by its membrane lipid composition and by the SV-resident membrane proteins.

Then, we assessed how differential syb2 D/M under different experimental conditions impacts the functional fusion pore as-sembly (Fig 8A). When the two donor membranes harboring similar syb2 monomers but significantly different dimers were compared, the membrane containing a reduced dimer population also showed a significant alteration in the individual fusion pores' properties (Fig 5). The t-SNARE densities in the acceptor membrane remained the same in both cases. The above ND5$_L$ pore properties can also originate because of syb2 dilution over the larger surface area of ND5$_L$, which does not trivialize the role of syb2 D/M in stabilizing the

**Figure 6. Syb2 TMD residues that control D/M also regulate pore properties.**
**(A)** Illustration shows the syb2 reconstituted in NDs (ND-syb2); syb2 TMD residues mutated in this set of studies are highlighted in the inset. **(B)** Representative immunoblot showing the presence of syb2 monomers (M) and dimers (D) in ND5$_S$ (WT and mutants as indicated), before (−) and after (+) the addition of t-lipo. The antibody used is mentioned. n = 5 (for WT), 8 (each for V101A and I105A), and 4 (for All ItoA). **(C)** Dimer-to-monomer band intensity ratio was normalized to WT syb2 reconstituted in ND5$_S$ without t-lipos (filled black), and is plotted as "Dimer: monomer ratio" (D/M). Individual data points corresponding to ND5$_S$ reconstituted with WT, V101A, I105A, and All ItoA syb2 mutants are represented as black, red, blue, and green, respectively. Data points without and with t-lipos are represented as "filled" and "empty" diamonds, respectively. n = 5 (for WT), 8 (each for V101A and I105A), and 4 (for All ItoA); the mean ± SEM are shown. **(D)** Lipid mixing (%) between t-lipo and fluorescently labeled ND5$_S$ reconstituted with WT and different syb2 mutants; plotted as a function of time (in seconds). n = 3 independent experiments; data are presented as the mean ± SEM. **(E)** Half-time ($t_{1/2}$) for all the traces from (E) is plotted. n = 3 independent experiments; individual data points are shown with the mean ± SEM. The t test was performed to compare the two means, **P < 0.01 and *P < 0.05. **(F)** Representative traces of single pores for ND5$_S$, ND5$_S$(All ItoA), ND5$_S$(V101A), and ND5$_S$(I105A) (top to bottom), as indicated. Three different epochs of the raw traces—beginning, middle, and end—are shown. Closed (C), open (O), and partial open (P) states are indicated with the respective currents. **(G)** Open dwell time histogram of pores, as indicated. n = 4 independent BLMs, each for ND5$_S$, ND5$_S$(All ItoA), ND5$_S$(V101A), and ND5$_S$(I105A) (left to right); three sets of NDs of each type were used.
Source data are available for this figure.

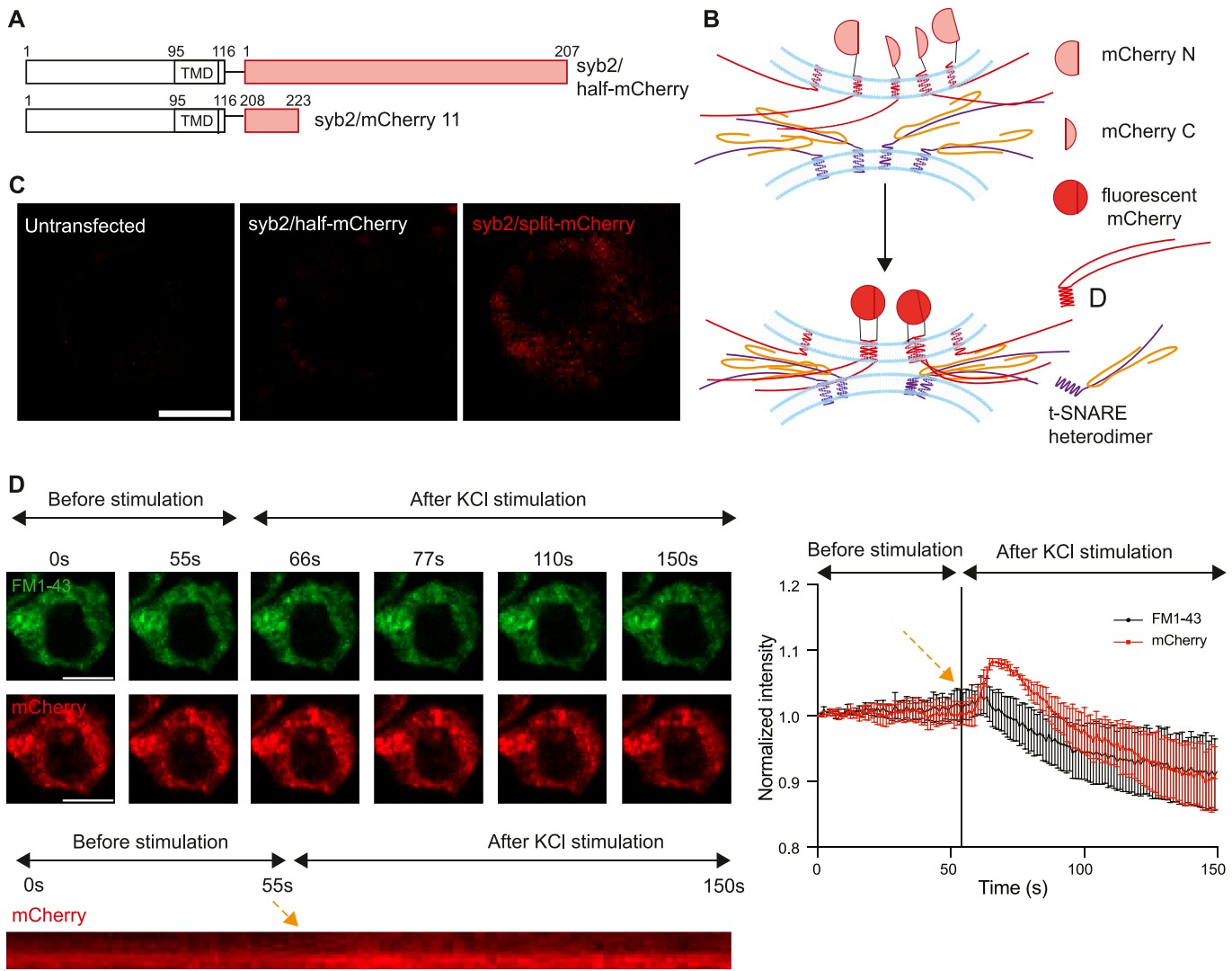

**Figure 7. Syb2 dimers are involved in stimulated secretion from PC12 cells.**
**(A)** Illustration shows the design of syb2/mCherry 11 and syb2/half-mCherry constructs. **(B)** Illustration of the BiFC approach used to visualize the formation of syb2 dimers in PC12 cells. mCherry fragments used in this assay are marked as mCherry N and mCherry C. **(C)** Representative images showing mCherry signal (561/610-nm ex/ em wavelength) in PC12 cells transfected with syb2/half-mCherry and syb2/split-mCherry constructs compared with untransfected cells (scale bar—5 μm). **(D)** Left top—Representative images of syb2/split-mCherry–transfected cells loaded with FM1-43 (488/570-nm ex/em wavelengths) imaged at different time points before and after 70 mM KCl stimulation (scale bar—5 μm). Left bottom, the resliced images of a mCherry region at the plasma membrane imaged during the live-cell experiment over 150 s (stimulation marked by the orange arrow). Right—normalized FM1-43 and mCherry signal intensity variations at the plasma membrane with time, before and after stimulation (marked by the orange arrow). n = 9, N = 3, where n is the number of cells analyzed, and N is the number of independent experiments. Data are represented as the mean ± SEM.

individual pores' open state. When the syb2 density (and the dimer population) was restored in ND55$_L$, the corresponding pores showed similar open-state stability as ND5$_S$.

We used a series of syb2 TMD mutations to point out the TMD residues that control syb2 D/M and also impact the fusion pore assembly (Fig 6). The syb2 All ItoA mutant's pore properties that we observed can also originate because of reduced TMD flexibility. Although previous studies pointed out an effect of reduced TMD flexibility in vesicular secretion from chromaffin cells (Dhara et al, 2016), it is not yet settled whether that has a direct impact on the individual pore properties. A single-point mutation in syb2 TMD that alters D/M also significantly altered the pore properties.

In contrast, single-point mutations in syb2 TMD that did not alter D/M did not show significant alteration in pore properties. Hence, the syb2 D/M might possess a regulatory role in stabilizing the fusion pore opening (Fig 8A and B).

Next, we wondered whether syb2 dimers accumulate at the plasma membrane during stimulated secretion from PC12 cells (Fig 7). Our syb2/split-mCherry construct indicates that the syb2 dimer population might express within PC12 cells. During the stimulated release, plasma membrane–localized dimer abundance increased in the plasma membrane. This increase was followed by the gradual decrease in the same over a long period, suggesting a slow dissolution of dimers over time. The parallel

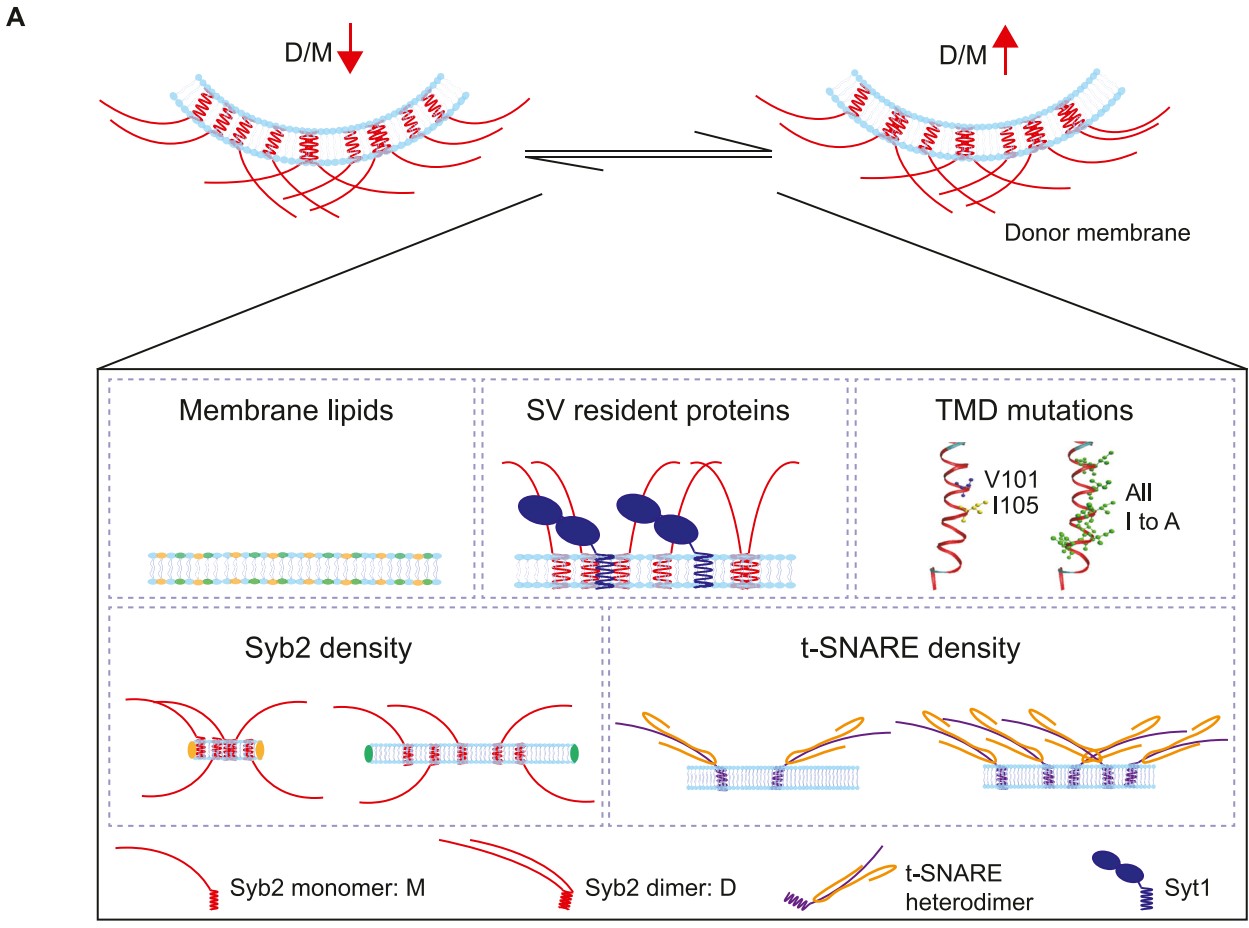

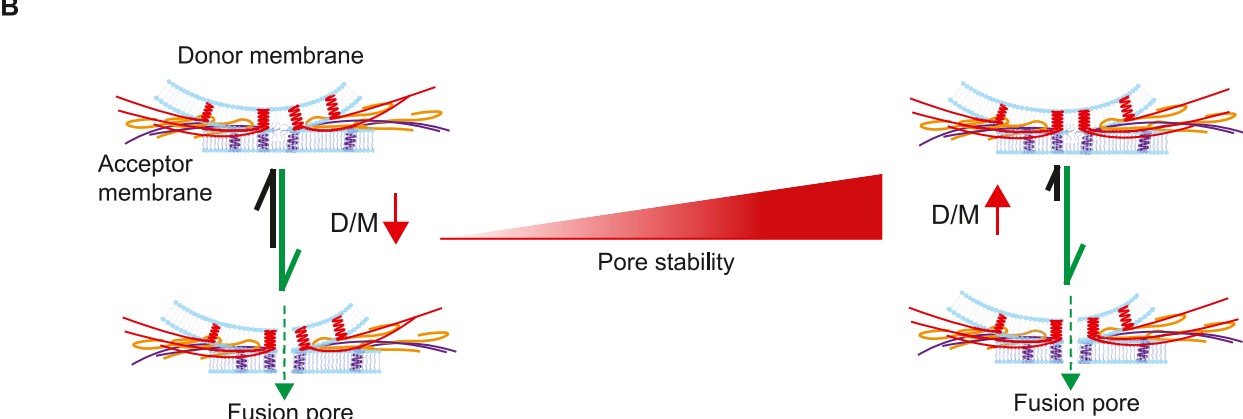

**Figure 8. Model describing syb2 monomer and dimer differential role in SNARE-mediated membrane fusion.**
**(A)** Illustration describes different factors controlling the syb2 D/M (D—dimers; M—monomers). Upward and downward red arrows indicate increased and decreased D/M, respectively. **(B)** Illustration describes the differential engagement of syb2 monomers and dimers in organizing the fusion pore assembly. Monomers and dimers are shown. Upward and downward red arrows indicate increased and decreased D/M, respectively.

FM 1–43 dye destaining (during stimulation) showed double exponential fluorescence decay kinetics, indicating the existence of multiple distinct modes of release, also proposed previously (Neher & Brose, 2018).

Overall, syb2 monomers and dimers control the v- and t-SNARE stoichiometry during the nascent fusion pore assembly. The syb2 dimer/monomer–mediated regulation of membrane fusion appears to have an evolutionary importance. Further studies are

required to get additional insight into how the syb2 monomers and dimers structurally reorganize during full fusion.

# Materials and Methods

### Animals

Sprague Dawley female rats (2.5–3 mo) were bred in the Tata Institute of Fundamental Research (TIFR) animal facility. The experimental procedures described below were in accordance with the guidelines of the Committee for Supervision and Care of Experimental Animals (CPCSEA), Government of India, and were approved by the TIFR Institutional Animal Ethics Committee (TIFR/IAEC/2020-4).

### Materials

1,2-Dioleoyl-sn-glycero-3-phospho-l-serine (PS), 1-palmitoyl-2-oleoyl-sn-glycero-3-phospho-(1′-rac-glycerol) (sodium salt) (PG), 1,2-dioleoyl-sn-glycero-3-phosphoethanolamine (PE), 1-palmitoyl-2-oleoyl-sn-glycero-3-phosphocholine (PC), 1,2-dioleoyl-sn-glycero-3-phosphoethanolamine-N-(lissamine rhodamine B sulfonyl) (Rhod-PE), 1,2-dioleoyl-sn-glycero-3-phosphoethanolamine-N-(7-nitro-2-1,3-benzoxadiazol-4-yl) (NBD-PE), and brain phosphatidylinositol 4,5-bisphosphate or PtdIns(4,5)P2 (PI[4,5]P2) were purchased from Avanti Polar Lipids; DDM (n-dodecyl $\beta$-D-maltoside) and OG (n-octyl glucoside) were from Gold Biotechnology; IPTG (isopropyl $\beta$-d-1-thiogalactopyranoside), Triton X-100, Hepes, KCl, imidazole, M$\beta$CD (methyl-$\beta$-cyclodextrin), and DSP (dithiobis succinimidyl propionate) were from Sigma-Aldrich; glutathione Sepharose 4 B and Ni Sepharose 6 Fast Flow were from GE Healthcare; $\beta$-mercaptoethanol and glycerol were from Thermo Fisher Scientific; Bio-beads SM2 were from BIO-RAD; BSA and protease inhibitor cocktail were from Roche; non-fat milk powder was from CST; Accudenz was from Axell; trypsin was from Thermo Fisher Scientific; cy3 and cy5 were from Cytiva. The antibodies used in this study are listed in Table S2.

### Protein purification

cDNA for the following proteins was derived from rats—neuronal SNAREs (syb2, syntaxin-1a) and syt1, whereas SNAP-25B was derived from mice (Das et al, 2020; Nellikka et al, 2021).

t-SNARE heterodimers (comprising syntaxin-1a and SNAP-25B) and syt1 were expressed as his-tagged proteins, at 37°C and 28°C, respectively, and purified, as described previously (Das et al, 2020; Nellikka et al, 2021). Bacterial pellets were resuspended (~10 ml per liter of culture) in resuspension buffer (25 mM Hepes–KOH, pH 7.4, 400 mM KCl, 10 mM imidazole, and 10 mM $\beta$-mercaptoethanol) containing protease inhibitor cocktail and DNase I. The samples were sonicated in 35-ml batches on ice for 6 × 15 s (20% duty cycle). Triton X-100 was added to 1% (vol/vol) and incubated overnight with rotation at 4°C before centrifugation of the cell lysate at 53,000$g$ for 30 min in a JA-25.5 rotor (Beckman). The supernatant was then incubated for >2 h at 4°C with Ni-NTA agarose (GE Healthcare; 0.5 ml of a 50% slurry per

liter of cell culture) equilibrated in resuspension buffer. Beads were washed extensively with resuspension buffer containing 1% Triton X-100 and then washed with OG wash buffer (25 mM Hepes–KOH, pH 7.4, 400 mM KCl, 10 mM imidazole, 10% glycerol, 5 mM $\beta$-mercaptoethanol, and 1% OG). The slurry was loaded onto a column, washed with 5–10 column volumes of OG wash buffer containing 50 mM imidazole, and step-eluted with OG wash buffer containing 500 mM imidazole. The purity of proteins was assessed using SDS–PAGE after staining with Coomassie brilliant blue.

Syb2 (WT and mutants as indicated in figures) was expressed as GST-tag protein at 37°C and purified using glutathione Sepharose beads, buffers, and other conditions described above. Purified syb2 from the beads was eluted by treating the beads with thrombin overnight, as described previously (Tucker et al, 2004).

MSPE3D1 and NW50 were also purified as his6-tagged proteins, as described previously (Nasr et al, 2017; Bao et al, 2018; Nellikka et al, 2021). In brief, a similar procedure as above was followed except all detergents were omitted from the wash buffers.

### Proteoliposome reconstitution

v- and t-SNARE liposomes were prepared as described previously (Shi et al, 2013; Nellikka et al, 2021). In brief, syb2 or t-SNARE heterodimers were mixed with lipids in reconstitution buffer (25 mM Hepes–KOH, pH 7.5, 100 mM KCl, and 2 mM $\beta$-mercaptoethanol) plus 0.3% OG. The following lipid compositions were used in different experiments—(1) 32% PE, 52% PC, 16% PS; (2) 31% PE, 52% PC, 16% PS, 1% PIP2; and (3) 75% PE, 25% PG. Dialysis was performed against reconstitution buffer (without OG), overnight at 4°C, followed by t-lipos' isolation by flotation using ACCUDENZ gradient. For this purpose, ultracentrifugation was performed at 180,000$g$ for 2 h in Optima MAX-XP ultracentrifuge (Beckman Coulter).

In case where t-SNARE copy numbers in the liposomes were varied, we used the following t-SNARE protein-to-lipid molar ratio—1.75:1,500 (t-lipo$_{[25]}$); 3.5:1,500 (t-lipo$_{[50]}$); 7:1500 (t-lipo$_{[100]}$); and 14:1500 (t-lipo$_{[200]}$). In the case of v-lipos, v-SNARE protein-to-lipid molar ratio was 10:1,500. When SVs were used in the experiments, the following lipid composition was used to prepare t-lipo—31% PE, 52% PC, 16% PS, 1% PIP2.

### ND reconstitution

Reconstitution of syb2 into NDs was performed as described previously (Shi et al, 2013; Nellikka et al, 2021). In some experiments, full-length syt1 was co-reconstituted with syb2 at a 1:2.5 (syt1:syb2) molar ratio, as described previously (Nellikka et al, 2021). MSP1E3D1 and NW50 were used to generate ~13 (ND$_S$) and ~50 nm (ND$_L$) NDs, respectively. The ratio of MSP to lipid molecules was 3:180 in the case of ND$_S$, whereas the MSP-to-syb2 ratios were 3:1.5 (ND3), 3:6 (ND5), and 3:9 (ND7). The ratio of NW50 to lipid molecules was 2:4,000 in the case of ND$_L$, whereas the NW50-to-syb2 ratios were 2:4 (ND5$_L$) and 2:33 (ND55$_L$). The syb2 and syt1 copy numbers per ND refer to the total number of syb2 and syt1 molecules (five copies of syb2 and 2 copies of syt1 per ND). The following lipid compositions were used as indicated: (1) 16% PS, 32% PE, 52% PC; (2) 16% PS, 29% PE, 52% PC, 1.5% NBD-PE, 1.5% Rhod-PE; (3) 100% PC; (4) 16% PS, 84% PC; (5) 32% PE, 68% PC; (6) 16% PS, 32% PE, 52% PC, 10% cholesterol; (7) 16% PS, 32% PE, 52% PC, 20%

cholesterol; (8) 16% PS, 32% PE, 52% PC, 30% cholesterol; and (9) 16% PS, 32% PE, 52% PC, 40% cholesterol. Briefly, reconstitution involved mixing syb2 (WT/mutants), MSP/NW50, and lipids with or without syt1 in a reconstitution buffer containing 0.02% DDM. The detergent was slowly removed by Bio-beads (1/3 volume) with gentle shaking (overnight, 4°C). The NDs in the supernatant were purified by gel filtration using a Superdex 200 10/300 GL column, equilibrated in reconstitution buffer. The ND samples were then concentrated using an Amicon filter unit (Merck) and stored at −80°C.

### Synaptic vesicle (SV) isolation

SVs from rat brains were isolated using the previously described protocol (Ahmed et al, 2013), with some modifications. In a tight-fitting glass–Teflon homogenizer (Remi), fresh rat brain tissue (~2.5 g) was homogenized in 9 ml ice-cold homogenization buffer (320 mM sucrose and 4 mM Hepes [pH:7.4], freshly supplemented with protease inhibitors, 0.001 volume). The homogenate was centrifuged at 1,000$g$ for 10 min at 4°C. The supernatant was collected and again centrifuged at 15,000$g$ for 15 min at 4°C. The SVs present in the supernatant were stored on ice. The pellet containing synaptosomes was split into two: one part was stored at −80°C, and the other part was further processed. To release the remaining SVs from synaptosomes, the pellet fraction was resuspended and homogenized by adding 9 ml of ice-cold homogenization buffer containing protease inhibitors (0.001 Vol/vol). The lysate was centrifuged at 17,000$g$ for 15 min at 4°C. The supernatant was collected and mixed with the previous supernatant containing SV, and this mixture was centrifuged at 48,000$g$ for 25 min at 4°C. Again, the supernatant was collected. To prevent SV aggregation, the supernatant was homogenized using 3–5 strokes (1,000$g$). The supernatant was drawn through a 20-gauge hypodermic needle and expelled through a 27-gauge needle; this was repeated twice. Then, 13 ml of this sample was overlaid onto 13 ml of 0.7 M sucrose cushion and centrifuged at 133,000$g$ for 1 h at 4°C using a 70 Ti rotor (Beckman Optima XPN-100 ultracentrifuge). After centrifugation, 1 ml fractions starting from the top of the gradient to the bottom were collected, and the pellet formed at the bottom was resuspended by adding 200 $\mu$l of homogenization buffer. To check the purity of the sample, a dot blot was performed (Fig S1) using monoclonal antibodies against proteasome components, PSMC6, Hsc70, and syb2. The fractions exclusively positive for syb2 were collected together (Fig S1). This mixture was centrifuged at 300,000$g$ for 2 h in a 70 Ti rotor (Beckman Optima XPN-100 ultracentrifuge), and SVs were pelleted. The supernatant was discarded, and the pellet (SV) was resuspended in ~200 $\mu$l of reconstitution buffer (25 mM Hepes–KOH, pH 7.5, 100 mM KCl, and 2 mM $\beta$-mercaptoethanol). Then, the resuspended SVs were again drawn through a 20-gauge hypodermic needle and expelled through a 27-gauge needle; this was repeated twice. SV samples were quantified by resolving in SDS–PAGE along with purified syb2 with known concentrations, and probed with the syb2 antibody (Fig S1). Finally, SVs were snap-frozen in liquid nitrogen and stored at −80°C.

### Immunoblotting

Immunoblotting was performed by mixing the samples with 1X (final concentration) Laemmli sample buffer and heated at 95°C for 5 min. Samples were run on SDS–PAGE and blotted onto a PVDF membrane. The membrane was blocked with 5% skimmed milk in Tris-buffered saline containing 0.1% Tween-20 (TBS-T) and incubated with respective primary antibodies (Table S2) overnight at 4°C. The blots were washed with TBS-T and incubated with the corresponding HRP-conjugated secondary antibody (Table S2) for 1 h at room temperature. The blots were developed using enhanced chemiluminescence (CST and Thermo Fisher Scientific) and imaged in an AI600 chemiluminescence imager (GE). For SNARE complex detection, samples were prepared without boiling.

### Probing for reconstituted proteins in NDs and liposomes

In the case where syb2 monomers or dimers were probed in NDs and v-lipos, the respective samples were incubated with or without t-lipos (generally, 1:1 for v- to t-lipo and 10:1 for ND to t-lipo were used; exceptions are mentioned in the figure legends) in reconstitution buffer (25 mM Hepes–KOH, pH-7.4, 100 mM KCl, and 2 mM $\beta$-mercaptoethanol), overnight at 4°C. The samples were probed with syb2 and syt1 antibodies (Table S2) (as indicated). The blots were analyzed using ImageJ software, and the amount of monomers and dimers was quantified by measuring the respective band intensities.

To probe SNARE complex formation in the above experiments, samples were probed with syb2 and SNAP-25B antibodies (Table S2) (as indicated). To further confirm the SNARE assembly, samples were heated at 95°C for 30 min.

### Probing for SV-resident proteins

SVs were incubated overnight with or without t-lipos (ratios as mentioned in the figure legends) in the reconstitution buffer (25 mM Hepes–KOH, pH-7.4, 100 mM KCl, 2 mM $\beta$-mercaptoethanol, 1 mM Ca$^{2+}$, and 0.5 mM BAPTA), at 4°C. The samples were probed with syb2, syt1, and syp antibodies in the immunoblots. The blots were analyzed using ImageJ software, and the amount of monomer, dimer, syt1, and syp was quantified by measuring the respective band intensities. In the case where we probed for SV-resident proteins, the blots were generally cut and probed separately, except Fig 1A.

To probe SNARE complex formation in the above experiments, samples were probed with the syb2 antibody (Table S2) (as indicated). To further confirm the SNARE assembly, samples were heated at 95°C for 30 min. In all the samples where SV was used to trace the SNARE complex assembly, 0.5 mM [Ca$^{2+}$]$_{free}$ was present in the reaction mixture.

In the case where M$\beta$CD was used, SVs were incubated with 10 mM M$\beta$CD for different duration: 0.5 h, 1 h, 1.5 h, and 2 h. The samples were probed with the syb2 antibody in the immunoblots. The blots were analyzed using ImageJ software, and the number of monomers and dimers was quantified by measuring the respective band intensities.

In the case where the DSP cross-linker was used, the manufacturer (Sigma-Aldrich) protocol was followed. Briefly, SVs were incubated with 2 mM of DSP (dissolved in DMSO) at RT for 30 min followed by 2-h incubation at 4°C. To stop the reaction, 50 mM Tris, pH-7.5, was added and incubated for an additional 15 min. The samples were probed with the syb2 antibody in the immunoblots. The blots were analyzed using ImageJ software, and the number of monomers and dimers was quantified by measuring the respective band intensities.

### Ensemble FRET assay

The purified syb2 was labeled site-specifically at C103 of TMD using CyDye maleimides (Amersham/GE Healthcare) following the protocol mentioned by the manufacturer. Briefly, the syb2 was given a buffer exchange with the conjugation buffer (25 mM Hepes–KOH, 150 mM KCl, 10% glycerol, and 1% OG) containing a 100 M excess of TCEP. The cy3 and cy5 dyes were added to the syb2 in the conjugation buffer at protein:dye molar ratios of 1:10, respectively. The cy3-labeled syb2 and the cy5-labeled syb2 (at C103), in a 2:1 M ratio, were reconstituted into ND and liposome, containing ~5 and ~150 copies of syb2. For the FRET assay, 5 nM fluorescent-labeled ND5$_S$/v-lipo was mixed with t-lipo in a 96-well flat black plate (Costar) and cy5 fluorescence was recorded with time in a SparkControl Magellan microplate reader (Tecan). The following parameters were set for the experiment in the plate reader: (1) excitation and emission wavelengths were set to 550 and 670 nm, respectively; and (2) temperature: 30°C.

In a separate set of experiments, cy3- and cy5-labeled NDs (ND5$_S^{cy3/cy5}$) were allowed to react with t-lipos (~200 copies) at 4°C overnight, in reconstitution buffer (25 mM Hepes–KOH, pH-7.4, 100 mM KCl, and 2 mM $\beta$-mercaptoethanol). The samples were then resolved in 10% SDS–PAGE, followed by fluorescence imaging at high sensitivity using a Typhoon TRIO variable mode imager (GE Healthcare). The following parameters were set during imaging—excitation wavelength: 532 nm; emission wavelength: 670 nm.

### Ensemble lipid mixing assay

A lipid mixing assay using labeled NDs and t-lipos was performed as described previously (Weber et al, 1998; Shi et al, 2013). Briefly, 5 $\mu$l of fluorescent-labeled NDs (using NBD-PE plus Rhod-PE) was mixed with 45 $\mu$l of t-lipos in a 96-well flat black plate (Costar) and NBD fluorescence was recorded with time in a SparkControl Magellan microplate reader (Tecan). The following parameters were set for the experiment in the plate reader: (1) excitation and emission wavelengths were set to 460 and 538 nm, respectively; (2) temperature: 30°C. Maximum lipid mixing was recorded by adding 5 $\mu$l of 5% Triton, and the NBD fluorescence was recorded for an additional 10 min. The raw fluorescence values were converted into % lipid mixing and plotted as a function of time.

### Ensemble glutamate release assay

A glutamate release assay using ND5$_S$ (with varying lipid compositions as indicated in Fig S9) and glutamate-entrapped t-lipos was performed as described previously (Bao et al, 2016; Nellikka et al,

2021). Here, 4 nM t-lipo was mixed with 150 nM NDs (as indicated) in a 96-well flat black plate with 1 $\mu$M iGluSnFR in the reaction buffer (25 mM Hepes, pH-7.4, 100 mM KCl, and 2 mM $\beta$-mercaptoethanol). Glutamate release from t-lipo was measured as an increase in iGluSnFR fluorescence in a SparkControl Magellan microplate reader (Tecan). The following parameters were set for the experiment in the plate reader: (1) excitation and emission wavelengths were set to 480 and 510 nm, respectively; (2) temperature: 30°C. Maximum glutamate release percentage was recorded by adding 5 $\mu$l of 5% Triton, and the iGluSnFR fluorescence was recorded for an additional 10 min. The raw fluorescence values were converted into % glutamate release and plotted as a function of time.

### Planar lipid bilayer electrophysiology

Planar lipid bilayer (BLM) electrophysiology experiments were performed using Planar Lipid Bilayer Workstation from Warner Instruments (Bao et al, 2018; Das et al, 2020; Nellikka et al, 2021). In brief, lipids (BLM lipid composition—32% PE, 16% PS, and 52% PC; at 30 mg/ml in n-decane) were first painted onto a 150-$\mu$m aperture in a 1-ml polystyrene cup (Warner Instruments) and dried for 15 min. After that, the aperture was immersed in 1 ml of 25 mM Hepes–KOH, pH 7.4, with 100 mM KCl in the cis and 10 mM KCl in the trans chamber. Silver–silver chloride electrodes were connected to each chamber. The lipid solution was gently re-applied to the hole until a conductance-blocking seal was formed, determined by capacitance. This process was repeated either with a brush or with an air bubble until the desired capacitance was achieved.

### Single fusion pore measurements

Single-pore measurements were performed using the methods described previously (Bao et al, 2018; Das et al, 2020; Nellikka et al, 2021). In brief, after the BLM formation, t-lipos (composed of PE and PG) were added to the cis chamber, which spontaneously fused with the planar bilayer, thus depositing the t-SNAREs into the BLM. t-SNAREs were reconstituted into BLMs, at a density of 0.4 molecules per $\mu$m$^2$ (Bao et al, 2018). Then, to form fusion pores, v-SNARE NDs (containing different syb2 densities, WT, and mutated syb2—as indicated) were added to the cis chamber. Pores formed within 5–30 min and could be monitored for >60 min. Currents were recorded using Bilayer Clamp Amplifier BC-535 (Warner Instrument) and a Digidata 1550B (with HumSilencer) acquisition system (Molecular Devices Corp.). Single-channel recordings were acquired at 10 kHz using pCLAMP 11 (Molecular Devices, LLC.) software and were filtered at 5 kHz using a multisection Bessel filter ($\Delta\psi \equiv \psi_{cis} - \psi_{trans}$ ($\psi_{trans} \equiv 0$ mV)). All recordings were conducted at room temperature. Pore formation and dynamics were studied at $\Delta\psi = 0$ mV.

### Analysis of single fusion pore unitary currents

Single-channel data were analyzed using Clampfit 10.7 (Molecular Devices, LLC.) and MS Origin 2019 (OriginLab). In all Figures showing BLM recordings, the representative traces were filtered at 1 kHz for display purposes.

Current histograms were plotted using Clampfit 10.7 and fitted with Gaussian functions. To calculate the open lifetime of individual

pores, 15-min recordings of individual traces were analyzed. Dwell times corresponding to the fully open and fully closed states were measured in individual records using the event detector in Clampfit 10.7. Open-state dwell time histograms were plotted using MS Origin 2019 (OriginLab).

### EM sample preparation and data collection

A 5 $\mu$l of undiluted and diluted (1:50) samples for SVs or NDs (ND5$_S$ and ND55$_L$) was spotted onto Formvar carbon–coated (150 mesh copper) grids for 5 min, followed by air drying under the light. Grids were washed with Milli-Q (10 drops) to remove excess stain and then kept under the light for air drying (for ~3 h). Finally, samples were imaged using a Tecnai 200 kV transmission electron microscope.

### MS sample preparation and data collection

ND5$_S$ was subjected to in-gel trypsin digestion (Pierce Trypsin Protease, MS Grade; Thermo Fisher Scientific), followed by (mass spectrometry) MS. Briefly, ND5$_S$ was mixed with 1X (final concentration) Laemmli sample buffer and heated at 95°C for 5 min. Samples were run on SDS–PAGE and stained with Coomassie. The gel was cut at the desired band position (for monomer ~ 17 kD, and dimer ~ 25–35 kD), destained, reduced, and alkylated with IAA (iodoacetic acid), and digested using trypsin (O/N at 4°C). A 5 $\mu$l of trypsin-digested sample (monomer/dimer) was mixed with HCCA (alpha-cyano-4-hydroxycinnamic acid) matrix (1:1 ratio) and spotted onto grid, dried, and subjected to mass spectrometry (ultra-fleXtreme MALDI-TOF-MS, Bruker). Mass spectra were acquired over an $m/z$ (mass/charge) range from 500 to 3,500. The experimental masses were compared with predicted masses obtained using PeptideMass/Cutter Tool (ExPASy) with the following parameters: cysteine treated with iodoacetic acid, [M+H]$^+$, trypsin digestion.

### Plasmid construct and cloning strategy

To visualize syb2 dimers in PC12 cells, the BiFC approach was used (Fdez et al, 2010), with modifications. In brief, syb2 along with 11 aa linker (L) (5′ SGGSGGTVGSR 3′) and mCherry 1–10 (one part of sfCherry2 1–207 aa) is cloned under CMV promoter (syb2/half-mCherry) (Fig S14). To this half-construct, IRES (internal ribosome entry site), syb2, 11 aa linker (L), and mCherry 11 (other part of sfCherry2 208–223 aa) elements are cloned in the sequence mentioned (syb2/split-mCherry) (Fig S14). Primers were designed using InFusion Cloning Primer Design Tool (Takara Bio). All the vectors and inserts were amplified using CloneAmp PCR Mix (Takara Bio) and ligated in the 2:1 insert:vector ratio using an InFusion kit (Takara Bio).

### Cell maintenance and transfection

Pheochromocytoma 12 cells (PC12) were procured from the National Centre for Cell Science, Pune. The cells were grown and maintained in F-12K Nutrient Mixture (Gibco), supplemented with 10% FBS (Gibco) and penicillin–streptomycin (Gibco) at 37°C and 5% CO$_2$. ~1.5 × 10$^5$ cells were seeded on 35-mm glass-bottom dishes coated with poly-D-lysine (Gibco). The cells were transfected with either syb2/split-mCherry or syb2/half-mCherry constructs in a reduced serum condition using Lipofectamine 3000 reagent (Thermo Fisher Scientific) for ~48 h at 37°C and 5% CO$_2$. The cells were then loaded with FM1-43 (1:150 dil) (Thermo Fisher Scientific) for 1 h at room temperature.

### Live-cell imaging and analysis

The glass-bottom dishes were imaged on an Olympus FV3000 confocal microscope equipped with a 100x 1.4NA oil objective lens. Images (320 × 320 pixels) were captured during the live-cell time-series experiments at a time interval of 1.13 s for 150 s. For the time-series experiments, cells loaded with FM1-43 were recorded before and after stimulation with 70 mM KCl solution (Sigma-Aldrich). The images were acquired using a GaAsP PMT detector at a zoom of 5x. The ROIs in the FM1-43 (488/570 nm ex/em wavelength)–loaded cells were chosen by marking the cell boundary. The mCherry (561/610 nm ex/em wavelength) signals corresponding to FM1-43 ROIs were analyzed. Signal intensities of the desired ROIs over time were measured using ImageJ and plotted on GraphPad Prism 10 software.

### Statistical analysis

The number of independent trials is provided in the figure legends, along with the statistical tests that were performed. Error bars represent the SEM.

## Supplementary Information

## Acknowledgements

We thank Das laboratory members for their suggestions and comments regarding this study. We thank Rohith K Nellikka, Sulekha Bhat, Laxmi Yadav, Shreedha Prabhu, and Boby KV for their technical assistance, and Dr. Shital Suryavanshi, Darshana Kapri, Praachi Tiwari, and Seema Shirolikar (TIFR) for technical assistance and animal house usage. The assistance from the EM facility and mass spectrometry facility (Geetanjali Dhotre) of TIFR, Mumbai, is acknowledged. This study was supported by a grant from the Department of Atomic Energy (Govt. of India) and TIFR, Mumbai, to D Das.

### Author Contributions

SS Patil: data curation, software, formal analysis, validation, investigation, methodology, and writing—original draft.
K Sanghrajka: data curation, software, formal analysis, validation, investigation, methodology, and writing—original draft, review, and editing.
M Sriram: data curation, software, formal analysis, validation, and methodology.
A Chakraborty: data curation, formal analysis, validation, investigation, and methodology.

S Majumdar: data curation and formal analysis.

BR Bhaskar: data curation and formal analysis.

D Das: conceptualization, resources, data curation, formal analysis, supervision, funding acquisition, investigation, project administration, and writing—original draft, review, and editing.

### Conflict of Interest Statement

The authors declare that they have no conflict of interest.

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
