## [Reviewer comments · Life Science Alliance]

Life Science Alliance

Synaptobrevin2 monomers and dimers differentially engage to regulate functional trans-SNARE assembly

Swapnali Patil, Kinjal Sanghrajka, Malavika Sriram, Aritra Chakraborty, Sougata Majumdar, Bhavya Bhaskar, and Debasis Das
DOI: <https://doi.org/10.26508/lsa.202402568>

Corresponding author(s): *Debasis Das, Tata Institute of Fundamental Research*

Review Timeline:

Submission Date:	2024-01-02
Editorial Decision:	2024-01-03
Revision Received:	2024-01-05
Editorial Decision:	2024-01-09
Revision Received:	2024-01-12
Accepted:	2024-01-12

Transaction Report:

Please note that the manuscript was previously reviewed at another journal and the reports were taken into account in the decision-making process at *Life Science Alliance*.

Reviewer #1 Review

Comments to the Authors (Required):

Patil et al employed a reconstituted in vitro vesicle fusion assay and tested the role of Syb2 dimer in the fusion pore open state stability. The authors observed the presence of Syb2 dimer that forms or is disassembled depending on the membrane lipids composition and the presence of t-SNARE proteins. The authors also generated mutant forms of Syb2 (in the transmembrane region) that do not form a dimer efficiently (V101A and I105A mutations). Consistent with the authors' proposal, the mutant forms of Syb2 displayed decreased fusion pore open state stability in an artificial assay. Therefore, the authors claim that the monomer vs. dimer form of Syb2 differentially regulates the stability of membrane fusion pores.

Major concerns:

1. In isolated synaptic vesicles, the presence of dimer form of Syb2 is not very convincing. Each synaptic vesicle contains ~70 copies of Syb2, and the ratio of dimer:monomer of Syb2 of Fig1c is a lot less than 1:70. Therefore, most synaptic vesicles do not have any dimer form of Syb2.
2. It is unclear why the authors are interested solely in the dimer. In Fig1b indicates the presence of possible trimer and tetramer of Syb2.
3. The lipid compositions of membranes are very stable except for the PIP2 concentrations of the inner leaflet of the plasma membrane. Therefore, in neurons the dimer form of Syb2 may not be formed as demonstrated in Fig4.
4. V101A and I105A mutations of Syb2 clearly decreased the dimer formation and displayed decreased stability of membrane fusion pores. However, V or I to A mutations in the transmembrane region will decrease the Van der Waals interaction between the membrane and the transmembrane region of Syb2. The decreased grab of synaptic vesicle via the transmembrane region of Syb2 can decrease the stability of membrane fusion pores. The authors generated L99W and C103W mutant forms of Syb2 that do not alter the dimer formation. These mutations will increase the Van der Waals interaction between the membrane and Syb2 transmembrane region. Therefore, testing these mutant forms of Syb2 will be good controls.
5. Most importantly, functional significance of these observations in the context of neurotransmitter release should be assessed using a physiologically and cell biologically relevant system.

Reviewer #2 Review

Comments to the Authors (Required):

Although regulated exocytosis has received considerable attention, our understanding of the fusion pore has remained relatively rudimentary. Numerous studies have examined the fusion pore in cells by amperometry and to a lesser extent by imaging, but in many cases the manipulations produce relatively modest effects on the prespike foot that can be difficult to interpret. The current study bypasses these limitations by using artificial membranes containing v- and t-SNAREs. On the one hand, the use of artificial membranes can yield information not relevant to the native process, but it can also be crucial in elucidating mechanism. The authors also use synaptic vesicles for a subset of the experiments, corroborating the findings in proteoliposomes and nanodiscs.

The study begins with relatively simple biochemical and biophysical experiments showing that the SNARE complex preferentially incorporates VAMP2 dimers rather than monomers. The results also indicate factors (phospholipid headgroups, cholesterol, other synaptic vesicle proteins, SNARE number and density) that influence dimerization. The effects are not huge but are substantial and the corroboration of SNARE density with complex formation makes a compelling case for the role of dimerization. Performed in vitro mostly with artificial membranes, these initial experiments would comprise an interesting but limited study. However, the authors extend the analysis to the fusion pore by electrophysiology, showing that factors influencing SNARE density (and

VAMP2 dimerization) promote pore opening and residues implicated by previous work in dimerization interfere with pore opening, increasing the impact of the work. The traces and compiled data are both very convincing.

In the end, VAMP2 dimerization has been characterized before and we still do not know exactly how dimerization of VAMP2 contributes to pore opening. A similar analysis of pore behavior by electrophysiology has also been performed by other groups, elucidating a role for SNARE number. This work takes the functional analysis further, providing more basic information about the mechanisms involved and the role of dimerization. The work is beautifully and thoughtfully done but the impact is less clear. What would more clearly elevate it would be a more physiologically relevant cell-based system that tests the key findings made in vitro, where the previous work was also conducted.

Reviewer #1 Review

Comments to the Authors (Required):

The authors have diligently addressed my earlier questions.

Reviewer #2 Review

Comments to the Authors (Required):

The authors have now added data suggesting the formation of synaptobrevin dimers in PC12 cells, but they have not provided the requested information about its effect on release or the fusion pore.

January 3, 2024

Re: Life Science Alliance manuscript #LSA-2024-02568-T

Dr. Debasis Das
Tata Institute of Fundamental Research
Biological Sciences
Homi Bhabha Road, Navy Nagar
Colaba
Mumbai, Maharashtra 400005
India

Dear Dr. Das,

Thank you for submitting your manuscript entitled "Synaptobrevin2 monomers and dimers differentially engage in priming functional trans-SNARE assembly" to Life Science Alliance. We invite you to re-submit the manuscript, revised to tone down claims made about the direct effect of synaptobrevin dimers on release or the fusion pore.

Thank you for this interesting contribution to Life Science Alliance. We are looking forward to receiving your revised manuscript.

Sincerely,

B. MANUSCRIPT ORGANIZATION AND FORMATTING:

January 3, 2024

Re: Life Science Alliance manuscript #LSA-2024-02568-T

Dr. Debasis Das
Tata Institute of Fundamental Research
Biological Sciences
Homi Bhabha Road, Navy Nagar
Colaba
Mumbai, Maharashtra 400005
India

Dear Dr. Das,

Thank you for submitting your manuscript entitled "Synaptobrevin2 monomers and dimers differentially engage in priming functional trans-SNARE assembly" to Life Science Alliance. We invite you to re-submit the manuscript, revised to tone down claims made about the direct effect of synaptobrevin dimers on release or the fusion pore.

>> Thank you for giving us the opportunity to revise our manuscript #LSA-2024-02568-T. The revised ms is now modified, indicating the potential role of synaptobrevin2 **dimer/monomer ratio** in regulating steps of membrane fusion. We hope that these modifications will be acceptable to publish our results in your journal.

Reviewer #1 (Comments to the Authors (Required)):

The authors have diligently addressed my earlier questions.

>> We appreciate the reviewer for all the insightful comments, which helped us revising the ms significantly.

Reviewer #2 (Comments to the Authors (Required)):

The authors have now added data suggesting the formation of synaptobrevin dimers in PC12 cells, but they have not provided the requested information about its effect on release or the fusion pore.

>> We thank the reviewer for all the insightful comments. In the revised manuscript, we have now modified the text, which indicates that the **synaptobrevin2 dimer/monomer ratio** is important in regulating various steps of membrane fusion.

Reviewer #1 (Comments to the Authors (Required)):

Patil et al employed a reconstituted in vitro vesicle fusion assay and tested the role of Syb2 dimer in the fusion pore open state stability. The authors observed the presence of Syb2 dimer that forms or is disassembled depending on the membrane lipids composition and the presence of t-SNARE proteins. The

authors also generated mutant forms of Syb2 (in the transmembrane region) that do not form a dimer efficiently (V101A and I to A mutations). Consistent with the authors' proposal, the mutant forms of Syb2 displayed decreased fusion pore open state stability in an artificial assay. Therefore, the authors claim that the monomer vs. dimer form of Syb2 differentially regulates the stability of membrane fusion pores.

Major concerns:

1. In isolated synaptic vesicles, the presence of dimer form of Syb2 is not very convincing. Each synaptic vesicle contains ~70 copies of Syb2, and the ratio of dimer:monomer of Syb2 of Fig1c is a lot less than 1:70. Therefore, most synaptic vesicles do not have any dimer form of Syb2.

>> We agree with the reviewer here, that the dimer population in SVs is significantly low than the monomers, which has been observed by us and others (PMID:7929121). However, the ratio of dimer:monomer of Syb2 in Fig1c that the reviewer has referred, seems incorrect. We have quantified the ratio as 0.065 (± 0.005), which is ~5 times more than 1:70 (which is 0.014). Hence, we conclude that the dimer population is, indeed, small in SVs, but not absent.

When we checked the dimer:monomer ratio in synaptosome preparations, it was 0.33 (± 0.04), ~5 fold more than in SVs. This new dataset is now included in the revised ms, also provided below for ready reference. Please note that the synaptosomes are defined as isolated synaptic terminals from neurons, hence we expect the presence of both the donor and acceptor membranes in the synaptosome preparations. Because the dimer:monomer ratio has altered significantly in synaptosome than SVs, we speculate that the presence of acceptor membrane (and perhaps the presence of t-SNAREs) is enhancing the syb2 dimer population in synaptosome fractions.

Additionally, we are now providing new set of cell-based data showing the enhancement followed by decrease of syb2 dimer population at the plasma membrane of PC12 cells, during stimulated vesicular secretion. Please see our response to reviewer #1, comment #5.

The result section now reads –

“We first probed the occurrence of syb2 monomers and dimers in the physiological environment (Fig. 1a). SVs isolated from adult rat brains (Fig. S1a-c) showed a detectable syb2 dimer population (Fig. 1a), which was significantly less compared to the monomers (Fig. 1a). The addition of cross-linker DSP (dithiobis succinimidyl propionate) enhanced the dimer population in SVs (Fig. S1d), also shown previously (Calakos and Scheller, 1994; Edelman et al., 1995). It indicates that syb2 monomers are indeed present nearby within SV, however, membrane lipids and/or SV resident protein(s) presumably prevent syb2 dimerization. Interestingly, dimer population exists significantly more in synaptosome fractions, in comparison to SVs, resulting in a significant increase of the dimer – to – monomer ratio (D/M) in synaptosome than SVs (Fig. 1a). This set of experiments indicated the existence of a heterogeneous syb2 D/M in the physiological environment, however, the functional consequence of this heterogeneity was unclear.”

The discussion section now reads –

“In our experiments, SVs harboured significantly less dimers in comparison to monomers. However, D/M increased ~5-fold in synaptosome fraction than SVs, indicating that dimers abundance can be heterogeneous in different physiological condition.”

a, left - Representative immunoblots showing the presence of syb2 monomers (M) and dimers (D) in synaptic vesicle (SV) isolated from rat brain. The antibody used is mentioned. $n = 10$ independent blots; $N = 4$ independent SV isolation. Middle - Representative immunoblots showing the presence of syb2 monomers (M) and dimers (D) in synaptosome fraction isolated from rat brain. The antibody used is mentioned. $n = 7$ independent blots; $N = 3$ independent synaptosome isolation. Right - Dimer to monomer band intensity ratio was plotted as "Dimer: monomer ratio" corresponding to SV (black) and synaptosome (red). $n = 7$ independent blots, $N = 3$ independent isolation; individual data points are shown along with mean \pm SEM. The Student's T-test was performed to compare the two means, $***p < 0.001$.

2. It is unclear why the authors are interested solely in the dimer. In Fig1b indicates the presence of possible trimer and tetramer of Syb2.

>> Please note - we have considered the other syb2 oligomers as well, shown in Fig. S5. We did not observe any significant change in the other syb2 oligomer population during SNARE complex assembly. Hence, we did not proceed with those higher order oligomers.

3. The lipid compositions of membranes are very stable except for the PIP2 concentrations of the inner leaflet of the plasma membrane. Therefore, in neurons the dimer form of Syb2 may not be formed as demonstrated in Fig4.

>> We agree with the reviewer that there will not be that broad level change in the lipid composition in the plasma membrane that is shown in Fig.4. However, the local lipid composition can have significant impact in the membrane fusion dynamics (PMID: 29037600, 18227127, 20513396, 32391794). To resolve the issue, we have now incorporated this section in the discussion of the current version of the ms. The relevant discussion section is provided below for the ready reference.

"The membrane lipid perturbations as demonstrated above might not be fully replicated inside the cell. However, the local lipid dynamics has been shown to impact membrane fusion (Churchward et al., 2008; Dhara et al., 2020; Kreutzberger et al., 2017; Zhang and Jackson, 2010). Our results indicate that the local lipid dynamics might impact the syb2 D/M, which has ramifications in fusion pore assembly."

4. V101A and I105A mutations of Syb2 clearly decreased the dimer formation and displayed decreased stability of membrane fusion pores. However, V or I to A mutations in the transmembrane region will decrease the Van der Waals interaction between the membrane and the transmembrane region of Syb2. The decreased grab of synaptic vesicle via the transmembrane region of Syb2 can decrease the stability of membrane fusion pores. The authors generated L99W and C103W mutant forms of Syb2 that do not alter the dimer formation. These mutations will increase the Van der Waals interaction between the membrane and Syb2 transmembrane region. Therefore, testing these mutant forms of Syb2 will be good controls.

>> We agree with the reviewer. To resolve the issue, we have now incorporated the single pore measurement data for L99W and C103W mutants in the current version of the ms. The relevant data and the text are provided below for ready reference.

The result section now reads –

“The V or I to A mutations in the syb2 TMD can decrease the Van der Waals interaction between the membrane and the TMD region of Syb2 and can contribute in altering the pore properties. To check that, we have used syb2 L99W and C103W mutants, this will increase the above Van der Waals interaction. These L99W and C103W mutants did not affect syb2 dimer population significantly. As shown in the Fig. S13, none of these mutants altered size or open state kinetic stability of individual pores, as evident from the current histograms and the open state dwell time distribution (Fig. S13). Hence the Van der Waals interaction did not affect the fusion pore properties in the case of V or I to A mutations in the syb2.”

The discussion section now reads –

“In contrast, single point mutations in syb2 TMD that did not alter D/M, did not show significant alteration of pore properties.”

Figure S13: Effect of syb2 TMD mutations on fusion pore regulation.

a, Representative traces of single pores for ND5_s (L99W)(top), and ND5_s (C103W)(bottom), as indicated. Three different epochs of the raw traces - beginning, middle, and end are shown. Closed (C), and open (O) states are indicated with the respective currents. **b**, Current histograms corresponding to the traces shown in **a**, (ND5_s (L99W) - top, and ND5_s (C103W) - bottom). Closed (C) and open (O) states are indicated. **c**,

Open dwell-time histograms of pores for ND5_s (L99W)(top), and ND5_s (C103W)(bottom), as indicated. $n = 3$ independent BLMs, two sets of NDs of each type were used.

5. Most importantly, functional significance of these observations in the context of neurotransmitter release should be assessed using a physiologically and cell biologically relevant system.

>> To resolve the issue, in the current version of the ms, we are now providing new set of cell-based data showing the enhancement followed by decrease of syb2 dimer population at the plasma membrane of KCl stimulated PC12 cells. We have also proposed a model based on our data, how the syb2 dimers might be relevant for vesicular secretion inside the cell. The relevant Figure and the text are provided below for the ready reference.

The result section now reads –

Syb2 dimers contribute to stimulated secretion from PC12 cells

To test if the syb2 dimers are involved in vesicular secretion from living cells, a BiFC (Bimolecular fluorescence complementation) assay (Hu et al., 2002; Kerppola, 2006) was performed. We transfected PC12 cells with a plasmid containing two syb2 molecules, each fused with one part of the split mCherry (Fdez et al., 2010; Feng et al., 2019) (syb2/split-mCherry) (Figs. 7a,b and S14). The mCherry expression inside the cell indicated the presence of syb2 dimers (Fig. 7c). When we removed one half of the split mCherry from the plasmid (syb2/ half-mCherry), no significant mCherry signal was detected (Figs. 7c, S14), further confirmed that the mCherry signal was indeed originating from syb2 dimer population expressing inside the PC12 cells.

To investigate whether syb2 dimers localize at the plasma membrane during secretion, we first marked the plasma membrane boundary of the syb2/split-mCherry expressing PC12 cells by exogenously adding FM1-43 dye (Klima and Foissner, 2008) (Fig. 7d). We investigated how fluorescence signals from FM1-43 and mCherry alter with time, before and after KCl stimulation. The contribution of photo bleaching was included in the quantification (Fig. S15). We observed a gradual decrease of FM1-43 signal upon KCl stimulation (Fig. 7d), an indication of vesicular secretion from PC12 cells (Gaffield and Betz, 2006). Interestingly, the corresponding mCherry signal increased briefly upon KCl stimulation followed by decrease with time (Fig. 7d). The FM1-43 signal decay followed a double exponential kinetics, with minor fast and major slow kinetic components (Table S2), indicating the presence of two kinetically distinct LDCV pools involved in KCl stimulated secretion. The concomitant mCherry signal increase followed by decrease suggested that new dimers accumulate at the plasma membrane upon KCl stimulation, which is critical for the slow and sustained release from PC12 cells. Presumably, the heterogeneous fusion pore structures are involved in fast and slow release events; dimers being integral part of those structures that triggers slow release.

The discussion section now reads –

“Next, we wondered how syb2 dimers impact vesicular secretion from living cells (Fig. 7). Our syb2/split-mCherry construct confirmed the existence of syb2 dimer population within PC12 cells. During stimulated release, plasma membrane localized syb2 dimers’ abundance rapidly increased with time indicating a localized accumulation of dimers at the release site. This increase was followed by the gradual decrease of dimers’ abundance over a long period, suggesting a slow dissolution of dimers over time. The parallel FM 1-43 dye destaining (during stimulation) showed a double exponential fluorescence decay kinetics, indicating the existence of multiple distinct modes of release, also proposed previously (Neher and Brose, 2018). Our results indicate that sustained and slow FM 1-43 dye unloading from the PC12 cells utilized the newly formed dimers (Fig. 8), as evident from the co-plot of the dimer level variation along with the FM 1-43 dye destaining. In the current experimental set up, we could not fully capture whether the fast release component also requires the direct involvement of syb2 dimers.

Overall, dimers play a significant functional role in organizing dynamic fusion pore assembly at the release site (Fig. 8). Further studies are required to get additional insight how the syb2 dimers structurally reorganize during full fusion, after constituting the nascent fusion pore assembly.”

Figure 7: Syb2 dimers are involved in stimulated secretion from PC12 cells.

a, Illustration shows the design of syb2/mCherry 11 and syb2/half-mCherry constructs. **b**, Illustration of BiFC approach used to visualize the formation of syb2 dimers in PC12 cells. mCherry fragments used in this assay are marked as mCherry N and mCherry C. **c**, Representative images showing mCherry signal (561/610 nm ex/em wavelength) in PC12 cells transfected with syb2/half-mCherry and syb2/split-mCherry constructs compared to untransfected cells (scale bar - 5 μ m). **d**, Left, top - Representative images of syb2/split-mCherry transfected cells loaded with FM1-43 (488/570 nm ex/em wavelengths) imaged at different time points before and after 70mM KCl stimulation (scale bar - 5 μ m). Left bottom, - the resliced images of a mCherry region at the plasma membrane imaged during the live cell experiment over 150s (stimulation marked by the orange arrow). Right - normalized FM1-43 and mCherry signal intensity variations at the plasma membrane with time, before and after stimulation (marked by the orange arrow). $n = 9$, $N = 3$ where n is the number of cells analyzed and N is the number of independent experiments. Data represented as mean \pm SEM.

Reviewer #2 (Comments to the Authors (Required)):

Although regulated exocytosis has received considerable attention, our understanding of the fusion pore has remained relatively rudimentary. Numerous studies have examined the fusion pore in cells by amperometry and to a lesser extent by imaging, but in many cases the manipulations produce relatively modest effects on the prespike foot that can be difficult to interpret. The current study bypasses these limitations by using artificial membranes containing v- and t-SNAREs. On the one hand, the use of artificial membranes can yield information not relevant to the native process, but it can also be crucial in elucidating mechanism. The authors also use synaptic vesicles for a subset of the experiments, corroborating the findings in proteoliposomes and nanodiscs.

The study begins with relatively simple biochemical and biophysical experiments showing that the SNARE complex preferentially incorporates VAMP2 dimers rather than monomers. The results also indicate factors (phospholipid headgroups, cholesterol, other synaptic vesicle proteins, SNARE number and density) that influence dimerization. The effects are not huge but are substantial and the corroboration of SNARE density with complex formation makes a compelling case for the role of dimerization. Performed in vitro mostly with artificial membranes, these initial experiments would comprise an interesting but limited study. However, the authors extend the analysis to the fusion pore by electrophysiology, showing that factors influencing SNARE density (and VAMP2 dimerization) promote pore opening and residues implicated by previous work in dimerization interfere with pore opening, increasing the impact of the work. The traces and compiled data are both very convincing.

In the end, VAMP2 dimerization has been characterized before and we still do not know exactly how dimerization of VAMP2 contributes to pore opening. A similar analysis of pore behavior by electrophysiology has also been performed by other groups, elucidating a role for SNARE number. This work takes the functional analysis further, providing more basic information about the mechanisms involved and the role of dimerization. The work is beautifully and thoughtfully done but the impact is less clear. What would more clearly elevate it would be a more physiologically relevant cell-based system that tests the key findings made in vitro, where the previous work was also conducted.

>> We appreciate the reviewer for the comments. To resolve the issue regarding the physiologically relevant assay, in the current version of the ms we have now incorporated new set of cell-based data showing the enhancement followed by decrease of syb2 dimer population inside the PC12 cells during stimulated vesicular secretion. We have also proposed a model based on our data, how the syb2 dimers might be relevant for vesicular secretion inside the cell. Please see our response to reviewer #1, comment #5.

January 9, 2024

RE: Life Science Alliance Manuscript #LSA-2024-02568-TR

Dr. Debasis Das
Tata Institute of Fundamental Research
Biological Sciences
Homi Bhabha Road, Navy Nagar
Colaba
Mumbai, Maharashtra 400005
India

Dear Dr. Das,

Thank you for submitting your revised manuscript entitled "Synaptobrevin2 monomers and dimers differentially engage to regulate functional trans-SNARE assembly". We would be happy to publish your paper in Life Science Alliance pending final revisions necessary to meet our formatting guidelines.

- please be sure that the authorship listing and order is correct
- please upload all figure files as individual ones, including the supplementary figure files
- please add ORCID ID for the corresponding author -- you should have received instructions on how to do so
- please use the [10 author names et al.] format in your references (i.e., limit the author names to the first 10)
- please add callouts for Figures 8A,B;S2A,B; S4A-C;S8A,B; S10A-C; S9A, B; S13A-C; S14A, B; S15A, B to your main manuscript text;
- please upload your Tables in editable .doc or excel format

A. FINAL FILES:

B. MANUSCRIPT ORGANIZATION AND FORMATTING:

Sincerely,

January 12, 2024

RE: Life Science Alliance Manuscript #LSA-2024-02568-TRR

Dr. Debasis Das
Tata Institute of Fundamental Research
Biological Sciences
Homi Bhabha Road, Navy Nagar
Colaba
Mumbai, Maharashtra 400005
India

Dear Dr. Das,

Thank you for submitting your Research Article entitled "Synaptobrevin2 monomers and dimers differentially engage to regulate functional trans-SNARE assembly". It is a pleasure to let you know that your manuscript is now accepted for publication in Life Science Alliance. Congratulations on this interesting work.

DISTRIBUTION OF MATERIALS:

Again, congratulations on a very nice paper. I hope you found the review process to be constructive and are pleased with how the manuscript was handled editorially. We look forward to future exciting submissions from your lab.

Sincerely,
